# VideoAgent: Self-Improving Video Generation

## Abstract

Video generation has been used to generate visual plans for controlling robotic systems. Given an image observation and a language instruction, previous work has generated video plans which are then converted to robot controls to be executed. However, a major bottleneck in leveraging video generation for control lies in the quality of the generated videos, which often suffer from hallucinatory content and unrealistic physics, resulting in low task success when control actions are extracted from the generated videos. While scaling up dataset and model size provides a partial solution, integrating external feedback is both natural and essential for grounding video generation in the real world. With this observation, we propose VideoAgent for self-improving generated video plans based on external feedback. Instead of directly executing the generated video plan, VideoAgent first refines the generated video plans using a novel procedure which we call *self-conditioning consistency*, utilizing feedback from a pretrained vision-language model (VLM). As the refined video plan is being executed, VideoAgent collects additional data from the environment to further improve video plan generation. Experiments in simulated robotic manipulation from MetaWorld and iTHOR show that VideoAgent drastically reduces hallucination, thereby boosting success rate of downstream manipulation tasks. We further illustrate that VideoAgent can effectively refine real-robot videos, providing an early indicator that robotics can be an effective tool in grounding video generation in the physical world.

## 1 Introduction

Large text-to-video models pretrained on internet-scale data have broad applications such as generating creative video content (Ho et al., 2022; Hong et al., 2022; Singer et al., 2022) and creating novel games (Bruce et al., 2024), animations (Wang et al., 2019), and movies (Zhu et al., 2023). Furthermore, recent work show that video generation can serve as simulators of the real-world (Yang et al., 2023b; Brooks et al., 2024), as well as policies with unified observation and action space (Du et al., 2024; Ko et al., 2023; Du et al., 2023). These recent applications of text-to-video generation models hold the great promise of internet-scale knowledge transfer (e.g., from generating human videos to generating robot videos), as well as paving the way to generalist agent (e.g., a single policy that can control multiple robots with different morphologies in different environments to perform diverse tasks).

Nevertheless, text-to-video models have only had limited success in downstream applications in reality. For instance, in video generation as policy (Du et al., 2024; Ko et al., 2023), when an observation image and a language instruction are given to a video generation model, generated videos often hallucinate (e.g., objects randomly appear or disappear) or violate physical laws (e.g., a robot hand going through an object) (Yang et al., 2023b; Brooks et al., 2024). Such hallucinations and unrealistic physics have led to low task success rate when generated videos are converted to control actions through inverse dynamics models, goal conditioned policies, or other action extraction mechanisms (Wen et al., 2023; Yang et al., 2024; Ajay et al., 2024).

While scaling up dataset and model size can be effective in reducing hallucination in large language models (LLMs) (Hoffmann et al., 2022), scaling is more difficult in video generation models. This is partially because language labels for videos are labor intensive to curate. Moreover, video generation has not converged to an architecture that is more favourable to scaling (Yang et al., 2024). Scaling aside, being able to incorporate external feedback to improve generation is one of the other most important breakthrough in LLMs (Ouyang et al., 2022b). It is therefore natural to wonder what kind of feedback is available for text-to-video models, and how we can incorporate these feedback to further improve the quality of the generated videos.

Figure 1: **The VideoAgent Framework.** VideoAgent first generates a video plan conditioned on an image observation and task description similar to (Du et al., 2023), and undergoes (1) iterative video refinement using feedback from a vision language model (VLM), (2) using the VLM to select the best refined video plan to convert to control actions through optical flow, and (3) executing the control actions in an environment and improving video generation using real-world feedback and additional data collected online.

To answer this question, we explore two types of feedback that are natural to acquire for video generation models, namely AI feedback from a vision-language model (VLM) and real-world execution feedback when generated videos are converted to motor controls. To utilize these feedback for self-improvement, we propose VideoAgent. Different from video generation as policy, which directly turns a generated video into control actions (Du et al., 2023; Ko et al., 2023), VideoAgent is trained to refine a generated video plan iteratively using feedback from a pretrained VLM. During inference, VideoAgent queries the VLM to select the best refined video plan, followed by execution of the plan in the environment. During online execution, VideoAgent observes whether the task was successfully completed and further improves the video generation model based on the execution feedback from the environment and additional data collected from the environment. The improvement to the generated video plan comes in two folds: First, we propose *self-conditioning consistency* for video diffusion model inspired by consistency models (Song et al., 2023; Heek et al., 2024), which enables low-quality samples from a video diffusion model to be further refined into high-quality samples. Second, when online access to the environment is available, VideoAgent executes the current video policy and collect additional successful trajectories to further finetune the video generation model on the successful trajectories. A visual illustration of VideoAgent is shown in Figure 1.

We first evaluate the performance of VideoAgent in two simulated robotics manipulation environments, Meta-World (Yu et al., 2020) and iTHOR (Kolve et al., 2017), and show that VideoAgent improves task success across all environments and tasks evaluated. VideoAgent can even improve the success rate of difficult tasks by as much as 4X. Next, we provide a thorough study on the effect of different components in VideoAgent, including different ways to prompt the VLM and different types of feedback from the VLM, providing a recipe for utilizing VLM feedback for video generation. Lastly, we illustrate that VideoAgent can iteratively improve real-robot videos, providing early signal that robotics can be an important mean to ground video generation models in the real world.

## 2 BACKGROUND

In this section, we provide the background on video generation as policy in a decision making process (Du et al., 2023). We also introduce consistent diffusion models (Song et al., 2023; Heek et al., 2024; Daras et al., 2024), which VideoAgent builds upon for self-refinement.

### 2.1 VIDEO AS POLICY IN SEQUENTIAL DECISION MAKING

We consider a predictive decision process similar to (Du et al., 2024): $\mathcal{P} := \langle \mathcal{X}, \mathcal{G}, \mathcal{A}, H, \mathcal{E}, \mathcal{R} \rangle$, where $\mathcal{X}$ denotes an image-based observation space, $\mathcal{G}$ denotes textual task description space, $\mathcal{A}$ denotes a low-level motor control action space, and $H \in \mathbb{R}$ denotes the horizon length. We denote $\pi(\cdot|x_0, g) : \mathcal{X} \times \mathcal{G} \mapsto \Delta(\mathcal{X}^H)$[1] as the language conditioned video generation policy, which models the probability distribution over $H$-step image sequences $\mathbf{x} = [x_0, ..., x_H]$ determined by the first frame $x_0$ and the task description $g$. Intuitively, $\mathbf{x} \sim \pi(\cdot|x_0, g)$ correspond to possible visual paths for completing a task $g$. Given a sampled video plan $\mathbf{x}$, one can use a learned mapping $\rho(\cdot|\mathbf{x}) : \mathcal{X}^H \mapsto \Delta(\mathcal{A}^H)$ to extract motor controls from generated videos through a goal-conditioned policy (Du et al., 2023), diffusion policy (Black et al., 2023), or dense correspondence (Ko et al., 2023). Once a sequence of motor controls $\mathbf{a} \in \mathcal{A}^H$ are extracted from the video, they are se-

---

[1]We use $\Delta(\cdot)$ to denote a probability simplex function

quentially executed in the environment $\mathcal{E}$, after which a final reward $\mathcal{R} : \mathcal{A}^H \mapsto \{0, 1\}$ is emitted representing whether the task was successfully completed. For simplicity, we only consider finite horizon, episodic tasks. Given a previously collected dataset of videos labeled with task descriptions $\mathcal{D} = \{(\mathbf{x}, g)\}$, one can leverage behavioral cloning (BC) (Pomerleau, 1988) to learn $\pi$ by minimizing

$$\mathcal{L}_{\text{BC}}(\pi) = \mathbb{E}_{(\mathbf{x}, g) \sim \mathcal{D}}[-\log \pi(\mathbf{x} | x_0, g)]. \tag{1}$$

Equation 1 can be viewed as maximizing the likelihood of the videos in $\mathcal{D}$ conditioned on the initial frame and task description.

## 2.2 CONSISTENCY MODELS

Diffusion models (Ho et al., 2020; Song et al., 2020b) have emerged as an important technique for data distribution modeling. During training, the model learns to map noisy data (at various noise levels) back to clean data in a single step. Concretely, let $x^{(0)}$ denote a clean image and $x^{(t)}$ denote the noisy image at noise level $t$, where $t \in [0, T]$, the training objective for a diffusion model $f_\theta(x^{(t)}, t)$ can be written as

$$\mathcal{L}_{\text{diffusion}}(\theta) = \mathbb{E}_{x^{(0)}, \epsilon, t}\left[\|f_\theta(x^{(t)}, t) - x^{(0)}\|^2\right], \tag{2}$$

where $\epsilon \in \mathcal{N}(0, I)$ is the added noise, and $x^{(t)} = \sqrt{\alpha_t} x^{(0)} + \sqrt{1 - \alpha_t}\epsilon$ where $\alpha_t$ are time-dependent noise levels. Although diffusion models have achieved high-quality image/video generation, they require hundreds or thousands of denoising steps during inference, which induces tremendous computational cost. To overcome the slow sampling speed of diffusion models, *consistency models* (Song et al., 2023; Song & Dhariwal, 2023) were initially proposed by enforcing a consistency loss across different noise levels, i.e.,

$$\mathcal{L}_{\text{consistency}}(\theta) = \mathbb{E}_{x^{(0)}, \epsilon, t_1, t_2}\left[\|f_\theta(x^{(t_1)}, t_1) - \texttt{stopgrad}\big(f_\theta(x^{(t_2)}, t_2)\big)\|^2\right], \tag{3}$$

which encourages the output of the single-step map between different noise levels to be similar. In fact, both the diffusion loss in Equation 2 and the consistency loss in Equation 3 can be understood as exploiting the structure of the denoising procedure which corresponds to an ordinary differential equation (ODE). Specifically, as introduced in (Song et al., 2023; 2020a), the backward denoising procedure of a diffusion model can be characterized by an ODE, i.e.,

$$\frac{\mathrm{d}x^{(t)}}{\mathrm{d}t} = -t \cdot s(x^{(t)}, t), \tag{4}$$

with $s(x^{(t)}, t)$ is some score function. During the entire path along $t \in (\epsilon, \infty]$, following this ODE should always maps $x^{(t)}$ to $x^{(0)}$. If we parametrize the model $f(x^{(t)}, t)$ as the simulation following the ODE governed by $s(x^{(t)}, t)$, we obtain the diffusion loss (2). Meanwhile, for all $t, t' \in (\epsilon, \infty]$, we have $f(x^{(t)}, t) = f(x^{(t')}, t')$ along the simulation path, which induces the consistency loss (3). Therefore, we can combine the diffusion loss and the consistency loss together for model training, i.e.,

$$\mathcal{L}(\theta) = \mathcal{L}_{\text{diffusion}}(\theta) + \lambda \cdot \mathcal{L}_{\text{consistency}}(\theta), \tag{5}$$

where $\lambda$ denotes consistency regularization hyperparameter across different noise levels.

## 3 VIDEO GENERATION AS AGENT

In this section, we introduce VideoAgent to improve video plan generation. Section 3.1 establishes a new concept for video diffusion models, termed *self-conditioning consistency*, which enables iterative refinement of video plans. In Section 3.2, we discuss how the video diffusion model trained with self-conditioning consistency can be utilized to *refine* generated video plans during inference. Finally, Section 3.3 explores how VideoAgent completes the self-improvement loop by collecting additional online data to further train the video generation and refinement model.

### 3.1 VIDEO REFINEMENT THROUGH SELF-CONDITIONING CONSISTENCY

We consider first-frame-and-language conditioned video generation following (Du et al., 2023; Ko et al., 2023), which generates a sequence of image frames to complete the task described by the language starting from the initial image. Generated videos often exhibit realistic segments (e.g., the beginning) alongside hallucinated segments (e.g., the end) (Yang et al., 2023b). Hence, while a video plan may not fully complete the specified task, the realistic segments provide a foundation for refinement, focusing on correcting hallucinations to produce a coherent, task-completing video. To leverage this partial progress, we propose a novel self-conditioning consistency mechanism. The

mechanism iteratively refines the generated video by retaining realistic portions while correcting inconsistencies in less accurate regions, transforming hallucinated segments into coherent task completions. This intuition underpins the design of the self-conditioning mechanism.

Let $\mathbf{x}^{(0)}$ be a ground truth video, and $\hat{\mathbf{x}}$ a generated sample from the diffusion model. We define a *self-conditioning consistency* model $\hat{f}_\theta(\hat{\mathbf{x}}, \mathbf{x}^{(t)}, t)$, which takes a generated video $\hat{\mathbf{x}}$ and a noisy version of the ground truth $\mathbf{x}^{(t)}$ as input to predict the clean video. This enables iterative refinement by conditioning on the previously generated sample, as shown in Figure 2.

We observe that self-conditioning is inspired by a reparameterization of the implicit ODE solver for Equation 4 (Song et al., 2020a; Lu et al., 2022; Zhang & Chen, 2022; Chen et al., 2022). For instance, Song et al. (2020a) considered the first-order ODE solver for Equation 4 following:

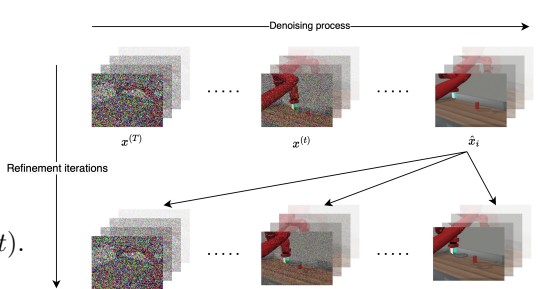

$$\mathbf{x}^{(t-1)} = \sqrt{\alpha_{t-1}}\mathbf{x}^{(0)} + \sqrt{1-\alpha_{t-1}-\sigma_t^2}\cdot s(\mathbf{x}^{(t)}, t). \tag{6}$$

In VideoAgent, we adapt Equation 6 by replacing the "predicted $\mathbf{x}^{(0)}$" term with $\hat{\mathbf{x}}$, the previously generated video sample, as illustrated in Figure 2. We remark that although $f(\cdot)$ shares similar parametrization to DDIM (Song et al., 2020a) as an ODE solver, our method uses $\hat{\mathbf{x}}$ from *previous* iterations, which is significantly

Figure 2: **An illustration of Self-Conditioning Consistency model.** Horizontal rows: denoising steps. Vertical rows: refinement iterations. $\hat{\mathbf{x}}_{i+1}$ denotes the generated video plan at refinement iteration $(i+1)$. We condition the refinement on the generated video from the previous iteration $\hat{\mathbf{x}}_i$.

different from traditional DDIM-based approaches that rely on "predicted $\mathbf{x}^{(0)}$" derived from $\mathbf{x}^{(t)}$ in the *same* iteration. This modification not only reduces the additional computational cost of DDIM for predicting $\mathbf{x}^{(0)}$, but more importantly, ensures alignment across iterations, progressively refining video quality through the self-conditioning mechanism. Introducing additional iteration-level consistency in the refinement procedure through the self-conditioning mechanism allows the denoising model to shortcut potential failures and reduce inference trials in the planning stage.

We learn the ODE solver through self-conditioning consistency by directly predicting the clean video $\mathbf{x}^{(0)}$ using:

$$\mathcal{L}_{\text{self-conditioning-consistency}}(\theta) = \mathbb{E}_{\hat{\mathbf{x}}, \mathbf{x}^{(0)}, t}\left[\|\hat{f}_\theta(\hat{\mathbf{x}}, \mathbf{x}^{(t)}, t) - \mathbf{x}^{(0)}\|^2\right]$$
$$+ \lambda \mathbb{E}_{\hat{\mathbf{x}}_1, \hat{\mathbf{x}}_2, t}\left[\|\hat{f}_\theta(\hat{\mathbf{x}}_1, \mathbf{x}^{(t)}, t) - \hat{f}_\theta(\hat{\mathbf{x}}_2, \mathbf{x}^{(t)}, t)\|^2\right]. \tag{7}$$

Equation 7 formalizes the self-conditioning consistency mechanism. The first term represents the standard diffusion loss, while the second term regularizes the similarity between independently generated samples ($\hat{\mathbf{x}}_1$ and $\hat{\mathbf{x}}_2$) to promote coherence across iterations. During training, we set $\lambda = 0$ to focus exclusively on individual sample consistency, while at inference, iterative refinement ensures alignment between subsequent predictions. This iterative refinement process distinguishes our approach from traditional consistency models, as we map any hallucinated or incoherent video to a progressively more realistic and coherent output. To enable the "first guess" for $\hat{\mathbf{x}}$, we consider $f_\theta(\mathbf{x}^{(t)}, t)$, which is still learned by the vanilla objective for video diffusion:

$$\mathcal{L}_{\text{video-diffusion}}(\theta) = \mathbb{E}_{\mathbf{x}^{(0)}, \epsilon, t}\left[\|f_\theta(\mathbf{x}^{(t)}, t) - \mathbf{x}^{(0)})\|^2\right]. \tag{8}$$

The overall objective for training a self-conditioning-consistent video diffusion model thus becomes:

$$\mathcal{L}(\theta) = \mathcal{L}_{\text{video-diffusion}}(\theta) + \lambda \mathcal{L}_{\text{self-conditioning-consistency}}(\theta). \tag{9}$$

Note that while the video generation model $f_\theta$ and the video refinement model $\hat{f}_\theta$ have different input arguments, we can share their parameters to train a single unified model for both video generation and refinement tasks. This parameter-sharing approach allows us to leverage the same model architecture for generating initial video plans and iteratively refining them using self-conditioning consistency. The training process for $f_\theta$ and $\hat{f}_\theta$ is detailed in Algorithm 1.

**Feedback Guided Self-Conditioning Consistency.** While we can refine videos only from previously generated samples, it may be desirable to condition the refinement process on any additional

feedback for the previously generated video that is available (e.g., feedback from humans or vision language models critiquing which part of the generated video is unrealistic). When such feedback is available, we can have the refinement model $\hat{f}$ further take the additional feedback as input, combined with the task description, to guide the refinement process, i.e.,

$$\hat{f}_\theta(\mathbf{x}, \mathbf{x}^{(t)}, t | \text{feedback}), \tag{10}$$

which can be plugged into our framework for learning using Equation 9.

## 3.2 INFERENCE THROUGH VLM GUIDED VIDEO GENERATION.

After training the video generation model $f_\theta$ and the video refinement model $\hat{f}_\theta$ described in Equation 8 and Equation 7, we can sample from $f_\theta$ and iteratively apply $\hat{f}_\theta$ for video refinement. Specifically, let $\eta$ be the step size for the noise schedule, $\sigma_t$ be a time dependent noise term, VideoAgent first "guesses" the video plan using the first-frame-and-language conditioned video generation generation, i.e.,

$$\mathbf{x}^{(t-1)} = \mathbf{x}^{(t)} - \eta \cdot \nabla_\theta f_\theta(\mathbf{x}^{(t)}, t) + \sigma_t \cdot \epsilon \tag{11}$$

The sample $\hat{\mathbf{x}}$ after $T$ denoising steps corresponds to the generated video. Next, we can iteratively apply $\hat{f}_\theta$ to refine the generated video sample

$$\hat{\mathbf{x}}_{(i+1)} = \hat{f}_\theta(\hat{\mathbf{x}}_{(i)}, \mathbf{x}^{(t)}, t), \tag{12}$$

where $i$ denotes the video refinement iteration, with $\hat{\mathbf{x}}_{(0)} = \hat{\mathbf{x}} = \mathbf{x}^{(T)}$. We denote the final video after refinement as $\hat{\mathbf{x}}_{\text{refined}}$. A natural question is when to stop the iterative video refinement process. One option is to always refine for a fixed number of iterations. However, over-refinement may lead to less diverse output. To overcome this, we leverage a VLM as a proxy for the environment's reward to assess whether a refined video is likely to lead to successful execution in the environment. Specifically, we denote a VLM as $\hat{\mathcal{R}}$, which takes a refined video $\hat{\mathbf{x}}_{(i)}$ and returns a binary value $\{0, 1\}$ to determine whether a video is acceptable based on overall coherence, adherence to physical laws, and task completion (See prompt for VLM in Appendix A). With $\hat{\mathcal{R}}$, the refinement stops when the VLM decides that the refined video is acceptable. Namely, we have

$$\hat{\mathbf{x}}_{\text{refined}} = \hat{\mathbf{x}}_{(i^*)}, \quad \text{where} \quad i^* = \min\left\{ i : \hat{\mathcal{R}}(\hat{\mathbf{x}}_{(i)}) = 1 \right\} \tag{13}$$

Algorithm 2 shows how video plans are generated, refined, and selected at inference time.

---

**Algorithm 1:** Training of Video Generation and Refinement Models with VLM Feedback

---

**Input:** Dataset $\mathcal{D}$, learning rate $\gamma$, total training iterations $N$, initial model parameters $\theta$, video generation model $f_\theta$, video refinement model $\hat{f}_\theta$, VLM $\hat{\mathcal{R}}$

**for** *iteration* $= 1$ **to** $N$ **do**

 Sample $\{(\mathbf{x}^{(0)}, g)\} \sim \mathcal{D}$ and $t \sim \text{Uniform}(\{0, 1, \ldots, T\})$;

 Compute vanilla diffusion loss:

  $\mathcal{L}_{\text{video-diffusion}} = \left\| f_\theta(\mathbf{x}^{(t)}, t) - \mathbf{x}^{(0)} \right\|^2$;

 Generate $\hat{\mathbf{x}}$ following Equation 11 and sample $\texttt{feedback} \sim \hat{\mathcal{R}}(\cdot | \hat{\mathbf{x}})$;

 Compute consistency loss:

  $\mathcal{L}_{\text{self-conditioning-consistency}} = \left\| \hat{f}_\theta(\hat{\mathbf{x}}, \mathbf{x}^{(t)}, t \,|\, \texttt{feedback}) - \mathbf{x}^{(0)} \right\|^2$;

 Update parameters:

  $\theta \leftarrow \theta - \gamma \nabla_\theta \left( \mathcal{L}_{\text{video-diffusion}} + \mathcal{L}_{\text{self-conditioning-consistency}} \right)$;

---

## 3.3 SELF-IMPROVEMENT THROUGH ONLINE FINETUNING

In addition to video refinement through self-conditioning consistency as described in Section 3.1, we can further characterize the combination of video generation and video refinement as a policy, which can be improved by training on additional real data collected from the environment during online interaction. Specifically, the goal is to maximize the expected returns of a policy through trial-and-error interaction with the environment:

$$\mathcal{J}_{\text{online}}(\theta) = \mathbb{E}\left[ \mathcal{R}(\mathbf{a}) \,|\, \pi_\theta, \rho, \mathcal{E} \right], \tag{14}$$

where $\mathcal{R}$ is the true reward function, $\mathcal{E}$ is the interactive environment, and $\pi_\theta$ corresponds to Algorithm 2, which contains both the video generation model $f_\theta$ and the video refinement model $\hat{f}_\theta$ as learnable components to be improved.

A broad array of reinforcement learning methods (Sutton & Barto, 2018) such as policy gradient (Schulman et al., 2017) can be employed to maximize the objective in Equation 14. For simplicity, we consider the setup of first executing the policy in the environment, then filtering for successful trajectories, continuing finetuning the video policy using additional online data, and executing the finetuned policy again to collect more data. Specifically, each online iteration constructs an additional dataset by rolling out the policy $\pi_\theta$ at the current online iteration

$$\mathcal{D}_{\text{new}} = \left\{ \hat{\mathbf{x}}_{\text{refined}} \sim \pi_\theta(x_0, g) \mid \mathcal{R}(\rho(\hat{\mathbf{x}}_{\text{refined}})) = 1 \right\}, \tag{15}$$

where $\rho$ is the optical flow model that maps the refined video to low-level control actions. See Algorithm 3 for details of online policy finetuning.

---

**Algorithm 2:** VLM Guided Replan

**Input:** Initial frame $x_0$, task description $g$,
    Reward $\mathcal{R}$, Environment $\mathcal{E}$, VLM $\hat{\mathcal{R}}$,
    max_refine_iterations, max_replans
**for** *replan_count = 1* **to** *max_replans* **do**
  $\hat{\mathbf{x}} \leftarrow \pi_\theta(x_0, g)$;
  **for** $i = 0$ **to** *max_refine_iterations* **do**
    response $\leftarrow \hat{\mathcal{R}}(\hat{\mathbf{x}}_{(i)}, g)$;
    **if** response $==$ ACCEPT **then break**;
    $\hat{\mathbf{x}}_{(i+1)} \leftarrow \pi_\theta(\hat{\mathbf{x}}_{(i)}, x_0, g)$;
  success $\leftarrow \mathcal{R}(\rho(\hat{\mathbf{x}}_{\text{refined}}))$;
  **if** success **then break**;
  $x_0 \leftarrow \mathcal{E}.\text{get\_state}()$;

---

**Algorithm 3:** Online Finetuning of Video Generation and Refinement Models

**Input:** Dataset $\mathcal{D}$, policy $\pi_\theta$, Reward $\mathcal{R}$,
    Environment $\mathcal{E}$
**for** *iteration $i = 1$ to $N$* **do**
  $\mathcal{D}_{\text{new}} \leftarrow \emptyset$;
  **for** *each $(\cdot, g)$ in $\mathcal{D}$* **do**
    $x_0 \leftarrow \mathcal{E}.\text{reset}(g)$;
    $\hat{x}_{\text{refined}} \sim \pi_\theta(x_0, g)$;
    **if** $\mathcal{R}(\rho(\hat{x}_{refined}))$ **then**
      $\mathcal{D}_{\text{new}} \leftarrow \mathcal{D}_{\text{new}} \cup (\hat{x}_{\text{refined}}, g)$;
  $\mathcal{D} \leftarrow \mathcal{D} \cup \mathcal{D}_{\text{new}}$;
  Finetune $\theta$ using Algorithm 1 on $\mathcal{D}$;

---

## 4 EXPERIMENTS

We now evaluate the performance of VideoAgent, introducing the experimental settings and variants of VideoAgent in Section 4.1, end-to-end success rate of VideoAgent against the baselines in Section 4.2 and the effect of different components of VideoAgent in Section 4.3. Finally, we show that VideoAgent is effective in improving the quality of real robotic videos in Section 4.4.

### 4.1 DATASETS AND EXPERIMENTAL SETUPS

**Datasets and Environments.** We follow the same evaluation setting as (Ko et al., 2023), which considers three datasets: Meta-World (Yu et al., 2020), iTHOR (Kolve et al., 2017), and BridgeData V2 (Walke et al., 2023). Meta-World consists of 11 robotic manipulation tasks performed by a simulated Sawyer arm, with video demonstrations captured from three distinct camera angles. iTHOR is a simulated 2D object navigation benchmark, where an agent searches for specified objects across four room types. BridgeData V2 is a real-world dataset of robotic manipulation. See more details of datasets and environments in Appendix C.

**Baselines and VideoAgent Variants.** We consider the following methods for comparison:

- **AVDC** (baseline). This is the Actions from Video Dense Correspondences (AVDC) (Ko et al., 2023) baseline, which synthesizes a video and predicts optical flow to infer actions.
- **AVDC-Replan** (baseline). When the movement stalls, AVDC-replan re-runs video generation and action extraction from the flow model to execute a new plan.
- **VideoAgent**. Our proposed video refinement model through self-conditioning consistency as introduced in Section 3.1. VideoAgent generates video and iteratively refines a video plan. We use GPT-4 Turbo for selecting the best video plan during inference (Section 3.2).
- **VideoAgent-Online**. As actions are executed in the online environment, successful trajectories are collected and used to continue training the video generation and refinement model, as described in Section 3.3.
- **VideoAgent-Online-Replan**. This variant incorporates online filtering of successful trajectories with the replanning mechanism, where replanning is conducted first, and more successful trajectories after replanning are added back to the training data.

### 4.2 END-TO-END TASK SUCCESS

**Meta-World.** We report the task success of baselines and VideoAgent in Table 1. Following Ko et al. (2023), we measure the average success across 3 camera poses with 25 seeds per pose. Without

Table 1: **Meta-World Results.** The mean success rates of baselines and VideoAgent on 11 simulated robot manipulation environments from Meta-World. VideoAgent consistently outperforms baselines across all tasks.

| | door-open | door-close | basketball | shelf-place | btn-press | btn-press-top |
|---|---|---|---|---|---|---|
| AVDC | 30.7% | 28.0% | 21.3% | 8.0% | 34.7% | 17.3% |
| AVDC-Replan | 72.0% | 89.3% | 37.3% | 18.7% | 60.0% | 24.0% |
| VideoAgent | 40.0% | 29.3% | 13.3% | 9.3% | 38.7% | 18.7% |
| VideoAgent-Online (Iter1) | 41.3% | 32.0% | 17.3% | 12.0% | 45.3% | 14.7% |
| VideoAgent-Online (Iter2) | 44.0% | 29.3% | 18.7% | 18.7% | 46.7% | 16.0% |
| VideoAgent-Online-Replan | **80.0%** | **97.3%** | **40.0%** | **22.7%** | **72.0%** | **40.0%** |

| | faucet-close | faucet-open | handle-press | hammer | assembly | **Overall** |
|---|---|---|---|---|---|---|
| AVDC | 12.0% | 17.3% | 41.3% | 0.0% | 5.3% | 19.6% |
| AVDC-Replan | 53.3% | 24.0% | 81.3% | 8.0% | 6.7% | 43.1% |
| VideoAgent | 46.7% | 12.0% | 36.0% | 0.0% | 1.3% | 22.3% |
| VideoAgent-Online (Iter1) | 38.7% | 13.3% | 36.0% | 0.0% | 4.0% | 23.2% |
| VideoAgent-Online (Iter2) | 49.3% | 21.3% | 44.0% | 1.3% | 1.3% | 26.4% |
| VideoAgent-Online-Replan | **58.7%** | **36.0%** | **85.3%** | **8.0%** | **10.7%** | **50.0%** |

online environment access, VideoAgent improves the overall success rate from self-conditioned consistency alone over the baseline (19.6% to 22.3%). Some tasks, such as faucet-close, show drastic improvement from 12% to 46.7%. With online data collection, VideoAgent-Online further improves success rates, with each online iteration (rolling out the policy, collecting successful trajectories, and continuing finetuning) boosting performance. When replanning is introduced, VideoAgent achieves 50% success, setting a new state-of-the-art. Detailed baseline results are in Appendix D.2, and qualitative improvements in refined videos are shown in Figure 9 in Appendix H.

**iThor.** Next, we evaluate VideoAgent on iThor. Due to the high computational cost of running the iThor simulator, we focus only on evaluating self-conditioning consistency (without online access). We follow the same setup as (Ko et al., 2023), where we measure the average success rate across four rooms each with three objects using 20 seeds. As shown in Table 2, VideoAgent consistently outperforms the baseline, demonstrating the effectiveness of self-conditioning consistency in producing more plausible video plans.

Table 2: **iThor Success Rates** comparing VideoAgent with the AVDC baseline.

| Room | AVDC | VideoAgent |
|---|---|---|
| Kitchen | 26.7% | **28.3%** |
| Living Room | 23.3% | **26.7%** |
| Bedroom | 38.3% | **41.7%** |
| Bathroom | 36.7% | **40.0%** |
| Overall | 31.3% | **34.2%** |

### 4.3 Understanding the Effect of Different Components in VideoAgent

In this section, we aim to understand the effect of different components of VideoAgent. Specifically, we focus on the effect of (1) different types of feedback given to the refinement model, (2) the number of refinement and online iterations, and (3) the quality of the VLM feedback.

#### 4.3.1 Effect of Different VLM Feedback.

In the previous section, we only used VLM during inference to determine when to stop refining a generated video. However, it is natural to wonder if information-rich feedback from the VLM, such as language descriptions of which part of a generated video to improve, might lead to better refined videos. To answer this question, we propose a few variants of VideoAgent according to the feedback available when training the video refinement model as in Equation 10. Specifically, we use VideoAgent to denote training the video refine-

Table 3: **Effect of Different Feedback** used to train the refinement model. Descriptive feedback from the VLM leads to higher improvement in task success.

| | **Overall** |
|---|---|
| AVDC | 19.6% |
| VideoAgent | 22.9% |
| VideoAgent-Binary | 23.8% |
| VideoAgent-Suggestive | 26.6% |

ment model only conditioned on the original task description. VideoAgent-Binary denotes additionally conditioning on whether a generated video is determined to be successful by the VLM. VideoAgent-Suggestive denotes conditioning additionally on language feedback from the VLM on which part of the video needs improvement and how the video can be improved. We train these three versions of the video refinement model, and report the overall task success from Meta-World in Table 3. We see that VideoAgent-Binary improves upon the base VideoAgent, while training with descriptive feedback in VideoAgent-Suggestive leads to even better performance. This sug-

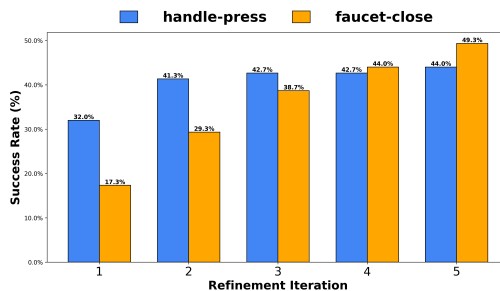
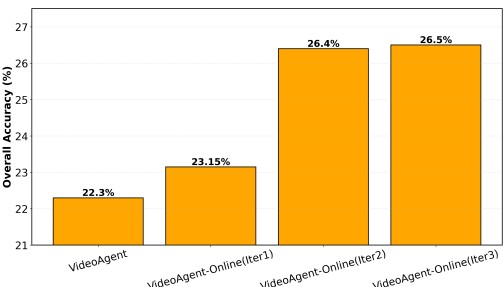

Figure 3: **Effect of Refinement Iterations.** The accuracy of downstream tasks generally increases as the number of refinement iteration increases.

Figure 4: **Effect of Online Iterations.** The overall task success of VideoAgent increases as the number of online iterations increases.

gests that richer feedback from the VLM can facilitate better training of the video refinement model. Improvement for each individual task can be found in the Appendix F.

### 4.3.2 Effect of Refinement and Online Iterations.

Next, we want to understand whether more refinement iterations and online finetuning iterations generally lead to higher task success. We found that while different tasks require a different number of iterations to achieve the best performance, VideoAgent does perform better as the number of refinement and online iterations increases, as shown in Figure 3 and Figure 4. During video refinement, specific tasks such as handle-press and faucet-close continue to see improvement even at the fifth refinement iteration. Faucet-close especially benefits from more refinement iterations, bringing success rate from 17.3% to 49.3% after five refinement iterations. The improved task success rates across refinement and online iterations suggests that self-conditioning consistency discussed in Section 3.1 and online interaction discussed in Section 3.3 can indeed effectively reduce hallucination and improve physical plausibility in the generated videos.

### 4.3.3 Accuracy of VLM feedback on Generated Videos.

Since this work is among the first to leverage a VLM to give feedback for video generation, it is crucial to understand whether a VLM can in fact achieve a reasonable accuracy in providing feedback for video generation. To quantify the performance of a VLM, we use human labels on whether a generated video is acceptable as the ground

Table 4: **VLM Performance** measured according to whether a VLM considers a generated video as acceptable using human label as the ground truth.

|  | Precision | Recall | F1-Score | Accuracy |
|---|---|---|---|---|
| **Unweighted** | 0.65 | 0.89 | 0.76 | 0.69 |
| **Weighted** | **0.92** | 0.58 | 0.71 | 0.75 |
| **Without Cam 3** | 0.91 | 0.71 | 0.80 | 0.79 |

truth, and measure precision, recall, F1-score, and accuracy based on whether GPT-4 Turbo thinks the generated video is acceptable according to trajectory smoothness (consistent across sequential frames), physical stability, and achieving the goal (See full prompt in Appendix A). We report the average result across 36 generated videos from the Meta-World dataset in Table 4. We see that the original prompt we used (Unweighted) achieves 69% accuracy, suggesting that the VLM is capable of judging generated videos. Since VideoAgent uses multiple refinement iterations, we want to avoid false positives where a bad video is accidentally accepted. We can achieve this by penalizing false positives through reweighting its cost in the prompt, which leads to the VLM rejecting videos when the VLM is uncertain about the video's acceptability. This adjustment results in a significant increase in precision as shown in Table 4. This weighted version of the prompt is used in the experiments in Section 4.2.

**Partial Observability.** In the AVDC experimental setup, center cropping the third camera (what is used in the pipeline) often results in most of the robot arm being outside of the frame. We found that the accuracy of the VLM is affected by such partial observatbility. As shown in Table 4, removing the third camera from the prompt leads to much higher accuracy.

**Descriptive Feedback.** While the VLM can provide binary feedback on whether a generated video is acceptable, we also measure the accuracy of the VLM in giving more descriptive feedback such as identifying the issue and providing suggestions on how to improve the video. We use three examples with human written language feedback as prompt for in-context learning. GPT-4 Turbo achieves 73.5% accuracy on identification and 86.1% accuracy on suggestion, as evaluated by humans. This

Table 5: **BridgeData-V2 Results.** Quantitative metrics comparing AVDC and VideoAgent on generated Bridge data. VideoAgent outperforms the baseline according to all except for one metric.

| Metrics | | AVDC | Video Agent |
|---|---|---|---|
| **Clip Score** | | 22.39 | **22.90** |
| **Flow Consistency** | | 2.48 ± 0.00 | **2.59 ± 0.01** |
| **Video Score** | Visual Quality | 1.97 ± 0.003 | **2.01 ± 0.003** |
| | Temporal Consistency | 1.48 ± 0.01 | **1.55 ± 0.01** |
| | Dynamic Degree | **3.08 ± 0.01** | 3.07 ± 0.02 |
| | Text to Video Alignment | 2.26 ± 0.003 | **2.30 ± 0.03** |
| | Factual Consistency | 2.02 ± 0.004 | **2.07 ± 0.01** |
| | **Average Video Score** | 2.16 ± 0.01 | **2.20 ± 0.01** |
| **Human Eval on Task Success** | | 42.0% | **64.0%** |

result is highly encouraging and opens up future directions of leveraging descriptive feedback from VLMs to improve video generation.

## 4.4 EVALUATING SELF-REFINEMENT ON REAL-WORLD VIDEOS

In this section, we evaluate VideoAgent's ability to refining real-world videos, which often contain higher variability, intricate details, nuanced behaviors, and complex interactions. We study the effect of video refinement using both quantitative metrics and qualitatively for holistic evaluation.

**Quantitative Evaluation.** Following previous literature on video generation, we consider two reference-free metrics, CLIP Score (Hessel et al., 2021) and Flow Consistency (Teed & Deng, 2020), as well as a set of Video-Scores (He et al., 2024). CLIP Score measures the cosine similarity between frame feature and text prompt, whereas Flow Consistency measure the smoothness and coherence of motion in the videos calculated from the RAFT model. Video-Scores use five sub-metrics with a focus on correlation with human evaluation and real-world videos.

We report the average across 2250 videos generated from the AVDC baseline and from VideoAgent in Table 5. VideoAgent performs better according to all metrics except for Dynamic Degree from Video-Score (which shows similar performance between the two methods). Notably, the gain is significant in metrics critical for real-world videos, such as CLIP Score, Factual Consistency, and Text-to-Video Alignment. Improvement in Flow Consistency and Temporal Consistency suggests that VideoAgent produces smoother and more physically plausible videos that adhere better to the physical constraints of the real-world. This directly translates to better performance in real-world robotic tasks in Table 1.

**Qualitative Evaluation.** Next, we qualitatively evaluate generated videos from the AVDC baseline and from VideoAgent. We collect 50 generated videos from each model and conduct human evaluation on whether a generated video looks realistic. Videos with refinement from VideoAgent improves the acceptance rate by 22% as shown in Table 5. We further show an example video with and without refinement in Figure 5, where the baseline (middle row) hallucinates (the bowl disappears) whereas VideoAgent produces the video that completes the task (bottom row). We also present a more fine-grained analysis of Visual Quality, Temporal Consistency, Dynamic Degree, Text to Video Alignment, and Factual Consistency evaluated by humans in the Appendix G with the metrics in Table 9, which further echos the results of human evaluations presented in Table 5.

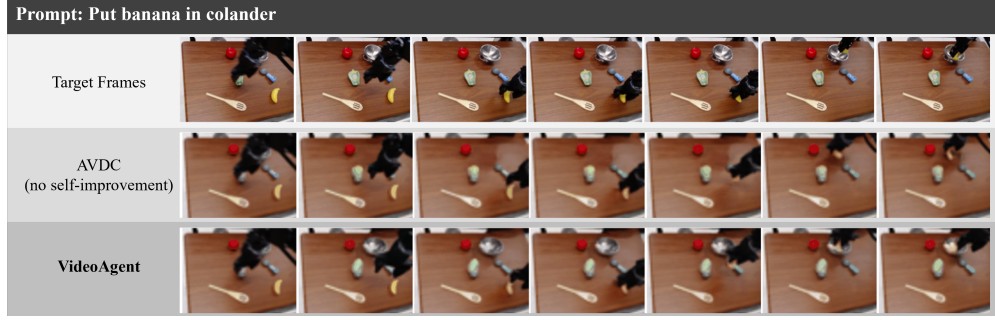

Figure 5: **Correcting Hallucinations in Video Generation:** The AVDC model hallucinates after the second frame, removing the colander and placing the banana on the table. In contrast, VideoAgent accurately retains the colander's position and correctly places the banana inside.

## 5 RELATED WORK

**Feedback and Self-improvement in LLMs.** Incorporating feedback and preference signals from feedback into the finetuning process of LLMs, has led to the enormous popularity and practical usability of the current versions of LLMs as chatbots (Casper et al., 2023). Preference feedback from humans or other AI systems (Ouyang et al., 2022a; Lee et al., 2023; Kaufmann et al., 2023) are first collected to train a reward model to guide the LLM's generation or do implicit policy optimization (Schulman et al., 2017; Rafailov et al., 2024). Furthermore LLMs have shown the ability to further improve by iterative refinement during finetuning and inference (Zelikman et al., 2022; Yuan et al., 2024; Tian et al., 2024). We incorporate this reward driven improvement mechanism in our work, but unlike the LLM setting where the feedback came from a reward model or some proxy of this prefernce model, in our VideoAgent we use natural feedback from real world when simulated videos are turned into actions that are executed in the real world.

**Image and Video Generation and Editing.** With the advent of large scale foundation models pretrained on internet scale data (Bommasani et al., 2021), generation of super realistic multimodal content has become easier. Text generation, image or video generation, and cross-modal generation (OpenAI et al., 2024; Reid et al., 2024; Wu et al., 2021; Ho et al., 2022; Singer et al., 2022; Yang et al., 2023a; Blattmann et al., 2023) has seen major advancements leveraging the autoregressive and diffusion based models architectures. And moving beyond simple generation, these models have been leveraged for guided text, image or video editing and enhancement (Huang et al., 2024) to improve textual and visual aesthetics applied mostly to generative media (Zhang et al., 2023). But none of these existing methods focus on grounding a generative simulator in the real world to perform more complex interactive multi-turn agentic and physical tasks needing both perception and control. To solve this bottleneck, we propose VideoAgent to self-improve or edit generated plan based on grounded feedback from real-world to execute robot manipulation tasks.

**Video Generation for Robot Learning.** Video-based learning for robotics has been extensively studied (Nair et al., 2022; Bahl et al., 2022; Shao et al., 2021; Chen et al., 2021; Pari et al., 2022; Sharma et al., 2019; Sun et al., 2018; Lee & Ryoo, 2017). Methods use video datasets for visual representation learning, goal extraction, and dynamic models for planning (Finn & Levine, 2017; Kurutach et al., 2018), or imitation learning from expert actions (Fang et al., 2019; Wang et al., 2023; Mani et al., 2024), state sequences (Torabi et al., 2019; Lee et al., 2021; Karnan et al., 2022), and pretraining on videos followed by RL (Baker et al., 2022; Escontrela et al., 2023). Recently, generative models have advanced video-based learning and planning, framing decision-making as text-conditioned video generation to predict trajectories (Du et al., 2024; Ko et al., 2023; Wen et al., 2023). Vision-language and text-to-video models generate long-horizon plans for robotic tasks through abstract, visually grounded planning (Du et al., 2023; Ajay et al., 2024). Generative models also simulate agent-environment interactions, enabling zero-shot deployment (Yang et al., 2023b), and test-time feedback for replanning (Bu et al., 2024). Unlike these, our VideoAgent improves video generation during training with real-world feedback and refines actions through test-time self-iteration and replanning.

## 6 CONCLUSION AND FUTURE WORK

We have presented VideoAgent, where a video generation model acts as an agent by generating and refining video plans, converting video plans into actions, executing the actions in an environment, and collecting additional data for further self improvement. Through interaction with an external environment, VideoAgent provides a promising direction for grounding video generation in the real world, thereby reducing hallucination and unrealistic physics in the generated videos according to real-world feedback. In order to fully achieve this overarching goal, VideoAgent needs to overcome a few limitations, which calls for future work:

- In the online setting, VideoAgent only considers filtering for successful trajectories for further finetuning. Exploring other algorithms such as online RL is interesting future work.
- VideoAgent utilizes optical flow for action extraction. It would be interesting to see how VideoAgent works with inverse dynamics model or image-goal conditioned diffusion policy.
- We only measured end-to-end task success in simulated robotic evaluation settings. It would be interesting to see how VideoAgent works with real robotic systems.
- As additional data is being collected in the online setting, in addition to finetuning the video prediction model, one can also finetune the action extraction module (flow model), and the VLM feedback model using the additionally collected data, which we defer to future work.

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

# A   PROMPT STRUCTURE FOR VLM FEEDBACK

## A.1   BINARY CLASSIFICATION

We employ a structured prompting strategy to provide feedback on video sequences for the zero-shot classification. The process consists of one Query-Evaluation Phase, each with distinct sub-goals.

---

### BINARY CLASSIFICATION

**Task:** You are a video reviewer evaluating a sequence of actions presented as seven consecutive image uploads, which together represent a single video. You are going to accept the video if it completes the task and the video is consistent without glitches.

**Query-Evaluation Phase:**

- **Inputs Provided:**
  - **Textual Prompt:** Describes the task the video should accomplish.
  - **Conditioning Image:** Sets the fixed aspects of the scene.
  - **Sequence of Images (7 Frames):** Represents consecutive moments in the video to be evaluated.

- **Evaluation Process:**
  - **View and Analyze Each Frame:** Examine each image in sequence to understand the progression and continuity of actions.
  - **Assess Overall Coherence:** Determine if actions transition smoothly and logically from one image to the next.
  - **Check for Physical Accuracy:** Ensure adherence to the laws of physics, identifying any discrepancies.
  - **Verify Task Completion:** Confirm the sequence accomplishes the task described in the textual prompt.
  - **Identify Inconsistencies:** Detect inconsistencies in object movement or overlaps that do not match the conditioning image.

- **Evaluation Criteria:**
  - Accept the sequence if it is a coherent video that completes the task.
  - Reject the sequence if any frame fails to meet the criteria, showing inconsistencies or not achieving the task. Be very strict, rejecting even minor errors.

- **Response Requirement:**
  - Provide a single-word answer: *Accept* or *Reject*. Do not give reasoning.

- **Additional Notes:**
  - No further clarification can be requested.
  - Elements from the conditioning image must match those in each frame of the sequence.

---

A.2 IDENTIFICATION AND SUGGESTION:

We employ a structured prompting strategy to provide descriptive feedback on video sequences via an in-context few-shot classification setup. The process consists of one Query-Evaluation Phase, each with distinct sub-goals.

---

### IDENTIFICATION AND SUGGESTION

**Task:** You are a video reviewer tasked with evaluating a series of actions depicted through eight consecutive image uploads. These images together simulate a video. This task is structured as a few-shot learning exercise, where you will first review three examples and then apply learned principles to new queries. **Query-Evaluation Phase:**

- **Inputs Provided:**
  - **Textual Prompt:** Describes the intended outcome or task the video aims to accomplish.
  - **Conditioning Image:** Establishes the fixed elements of the scene.
  - **Sequence of Images (7 Frames):** Illustrates consecutive moments in the video, representing the action sequence.

- **Evaluation Process:**
  - **Frame-by-Frame Analysis:** Carefully examine each of the seven images to understand the progression and continuity of actions.
  - **Assess Overall Coherence:** Evaluate the sequence as a whole to determine if the actions transition smoothly from one frame to the next while maintaining logical progression.
  - **Check for Physical Accuracy:** Ensure each frame complies with the laws of physics, identifying any discrepancies in movement or positioning.
  - **Verify Task Completion:** Confirm if the sequence as a whole accomplishes the task described in the textual prompt.
  - **Identify Inconsistencies:** Detect inconsistencies in object movement or overlaps that contradict the fixed scene elements depicted in the conditioning image.

- **Evaluation Criteria:**
  - **Descriptive Feedback:** Based on your evaluation, provide a concise, constructive sentence suggesting specific improvements. Focus on enhancing physical accuracy and task fulfillment based on identified inconsistencies or discrepancies.

- **Response Requirement:**
  - Feedback must be derived from your observations during the evaluation and not exceed 20 words.

- **Additional Notes:**
  - No further clarification can be requested.
  - Elements from the conditioning image must match those in each frame of the sequence.

---

## B   TASK DESCRIPTIONS AND IN-CONTEXT EXAMPLES FOR VLM FEEDBACK

### TASK DESCRIPTION AND SUCCESS CRITERIA

- **door-open**: The robot arm has to open the door by using the door handle.
- **door-close**: The robot arm has to close the door by pushing the door or the handle.
- **basketball**: The robot arm has to pick up the basketball and take it above the hoop.
- **shelf-place**: The robot arm has to pick up the blue cube and place it on the shelf.
- **button-press**: The robot arm has to press the red button from the side by pushing it inside.
- **button-press-topdown**: The robot arm has to press the red button from the top by pushing it downward.
- **faucet-close**: The robot arm has to use the red faucet handle and turn it anti-clockwise.
- **faucet-open**: The robot arm has to use the red faucet handle and turn it clockwise.
- **handle-press**: The robot arm has to press the red handle downward.
- **hammer**: The robot arm has to grip and pick up the hammer with a red handle and hit the peg on the box inside.
- **assembly**: The robot arm has to pick up the ring and place it into the red peg.

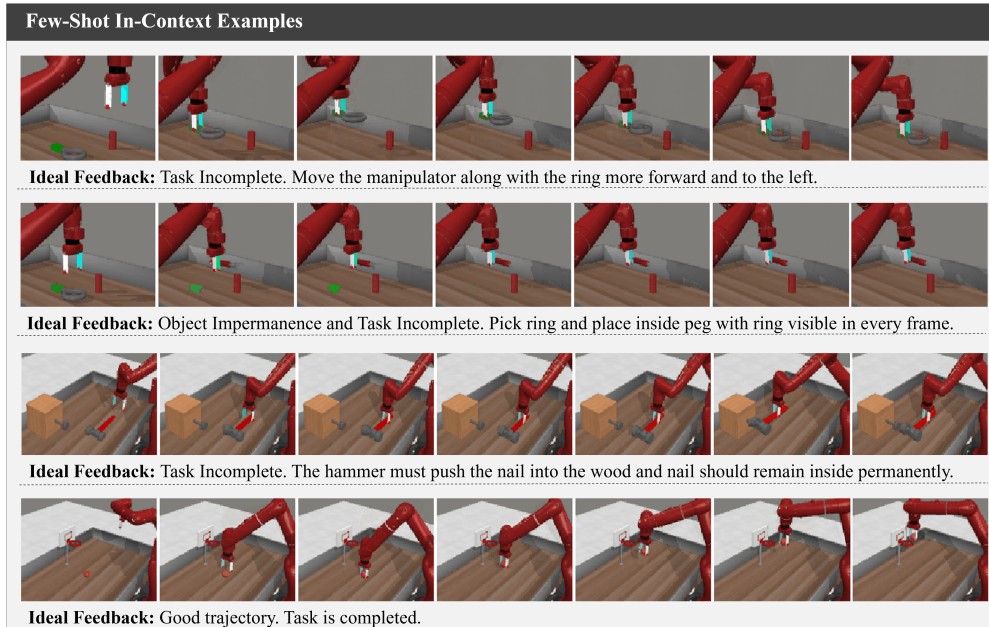

**Few-Shot In-Context Examples**

**Ideal Feedback:** Task Incomplete. Move the manipulator along with the ring more forward and to the left.

**Ideal Feedback:** Object Impermanence and Task Incomplete. Pick ring and place inside peg with ring visible in every frame.

**Ideal Feedback:** Task Incomplete. The hammer must push the nail into the wood and nail should remain inside permanently.

**Ideal Feedback:** Good trajectory. Task is completed.

Figure 6: Few-Shot Examples given to VLM: We provide some examples to the VLM and corresponding feedback to teach the VLM in-context how to critic the generated videos for task completion and success or failure.

## C   DATASET DESCRIPTIONS IN DETAIL

Meta-World (Yu et al., 2020) is a simulation benchmark that uses a Swayer robotic arm to perform a number of manipulation tasks. In our experiments, we make use of 11 tasks as shown in Table 1. We capture videos from three distinct camera angles for each task and use the same camera angles for both the training and testing phases. We gather five demonstration videos per task for each camera angle. During the evaluation, we tested on each of the three camera angles with 25 seeds per camera angle. The position of the robot arm and the object is randomized at the beginning of each seed to ensure variability. A trajectory is considered successful if the Video Agent reaches within a really close threshold of the goal state.

Figure 7: Environments and Datasets that we work with: Meta-World, iThor, and BridgeData-V2

iTHOR (Kolve et al., 2017) is another popular 2D simulated benchmark that focuses on embodied common sense reasoning. We evaluate the Video as Agent framework on the object navigation tasks, where an agent is randomly initialized in a scene and tasked with finding an object of a specified type (e.g., toaster, television). At each time step, the agent can take one of the four possible actions (MoveForward, RotateLeft, RotateRight, or Done), and observes a 2D scene to operate in. We selected 12 objects ((e.g. toaster, television) to be placed in 4 different room types (e.g. kitchen, living room, bedroom, and bathroom). Again, the starting position of the agent is randomized at the start of each episode. During evaluation, we test the agent across 12 object navigation tasks spread across all 4 room types, 3 tasks per room. A trajectory is successful if the agent views and reaches within 1.5 meters of the target object before reaching the maximum environment step or predicting Done.

To test the usefulness of our framework across different videos types, we also use the BridgeData V2 dataset (Walke et al., 2023), a large and diverse dataset of real world robotic manipulation behaviors designed to facilitate research in scalable robot learning. It contains 60,096 trajectories collected across 24 environments using a publicly available low-cost WidowX 250 6DOF robot arm. The dataset provides extensive task and environment variability, enabling skills learned from the data to generalize across environments and domains.

## C.1 ADDITIONAL TRAJECTORIES PER ITERATION DURING ONLINE TRAINING

We collect 15 successful trajectories for each task during every iteration. This standardization helps address task imbalance, as task success rates are higher for certain tasks compared to others. By ensuring a fixed number of successful trajectories per task, we prevent overfitting to easier tasks and maintain balanced model performance across the entire task set.

## D EXTENDED EXPERIMENTS

### D.1 VIDEOS TO ACTION CONVERSION

We employ the GMFlow optical flow model to predict dense pixel movements across frames. These predicted flows serve as the foundation for reconstructing both object movements and robot motions depicted in the video. The flow predictions allow us to interpret the temporal evolution of the video in terms of actionable physical dynamics. The optical flow essentially provides a dense correspondence of pixel movements between consecutive frames, which is then used to infer the relative motion of objects and the robot. This mapping bridges the gap between the high-dimensional video representation and the low-level control commands required to execute the tasks in a simulated or real environment.

This method ensures that the generated video plans are actionable and aligned with the task-specific dynamics, making the video generation process directly relevant to downstream policy learning and execution.

### D.2 BASELINE EXPERIMENTS ON METAWORLD

We conduct experiments on additional baselines including, Behavioral Cloning (BC), UniPi (with replan), VLP and Diffusion policy. Table 6 consists of these results.

### D.3 FURTHER ANALYSIS OF VIDEOAGENT-ONLINE

We train VideoAgent-Online for multiple iterations and observe that after 2 iterations, the results start to stabilize. The results for iteration 3 are shown in table 7.

Table 6: **Meta-World Results.** The mean success rates of baselines and VideoAgent on 11 simulated robot manipulation environments from Meta-World. VideoAgent consistently outperforms baselines across all tasks.

| | door-open | door-close | basketball | shelf-place | btn-press | btn-press-top |
|---|---|---|---|---|---|---|
| BC-Scratch | 21.3% | 36.0% | 0.0% | 0.0% | 34.7% | 12.0% |
| BC-R3M | 1.3% | 58.7% | 0.0% | 0.0% | 36.0% | 4.0% |
| UniPi (with Replan) | 0.0% | 36.0% | 0.0% | 0.0% | 6.7% | 0.0% |
| AVDC | 30.7% | 28.0% | 21.3% | 8.0% | 34.7% | 17.3% |
| VLP | 33.3% | 28.0% | 17.3% | 8.0% | 36.0% | 18.7% |
| Diffusion Policy | 45.3% | 45.3% | 8.0% | 0.0% | 40.0% | 18.7% |
| AVDC-Replan | 72.0% | 89.3% | 37.3% | 18.7% | 60.0% | 24.0% |
| VideoAgent | 40.0% | 29.3% | 13.3% | 9.3% | 38.7% | 18.7% |
| VideoAgent-Online (Iter1) | 41.3% | 32.0% | 17.3% | 12.0% | 45.3% | 14.7% |
| VideoAgent-Online (Iter2) | 44.0% | 29.3% | 18.7% | 18.7% | 46.7% | 16.0% |
| VideoAgent-Online-Replan | **80.0%** | **97.3%** | **40.0%** | **22.7%** | **72.0%** | **40.0%** |

| | faucet-close | faucet-open | handle-press | hammer | assembly | **Overall** |
|---|---|---|---|---|---|---|
| BC-Scratch | 18.7% | 22.7% | 28.0% | 0.0% | 0.0% | 15.4% |
| BC-R3M | 18.7% | 17.3% | 37.3% | 0.0% | 1.3% | 16.2% |
| UniPi (with Replan) | 4.0% | 9.3% | 13.3% | 4.0% | 0.0% | 6.1% |
| AVDC | 12.0% | 17.3% | 41.3% | 0.0% | 5.3% | 19.6% |
| VLP | 30.7% | 10.7% | 33.3% | 0.0% | 1.3% | 19.8% |
| Diffusion Policy | 22.7% | **58.7%** | 21.3% | 4.0% | 1.3% | 24.1% |
| AVDC-Replan | 53.3% | 24.0% | 81.3% | 8.0% | 6.7% | 43.1% |
| VideoAgent | 46.7% | 12.0% | 36.0% | 0.0% | 1.3% | 22.3% |
| VideoAgent-Online (Iter1) | 38.7% | 13.3% | 36.0% | 0.0% | 4.0% | 23.2% |
| VideoAgent-Online (Iter2) | 49.3% | 21.3% | 44.0% | 1.3% | 1.3% | 26.4% |
| VideoAgent-Online-Replan | **58.7%** | 36.0% | **85.3%** | 8.0% | **10.7%** | **50.0%** |

| | door-open | door-close | basketball | shelf-place | btn-press | btn-press-top |
|---|---|---|---|---|---|---|
| VideoAgent | 40.0% | 29.3% | 13.3% | 9.3% | 38.7% | 18.7% |
| VideoAgent-Online(Iter1) | 41.3% | 32.0% | 17.3% | 12.0% | 45.3% | 14.7% |
| VideoAgent-Online(Iter2) | 44.0% | 29.3% | 18.7% | 18.7% | 46.7% | 16.0% |
| VideoAgent-Online(Iter3) | 46.7% | 28.0% | 18.7% | 18.7% | 45.3% | 20.0% |

| | faucet-close | faucet-open | handle-press | hammer | assembly | **Overall** |
|---|---|---|---|---|---|---|
| VideoAgent | 46.7% | 12.0% | 36.0% | 0.00% | 1.3% | 22.3% |
| VideoAgent-Online(Iter1) | 38.7% | 13.3% | 36.0% | 0.00% | 4.0% | 23.15% |
| VideoAgent-Online(Iter2) | 49.3% | 21.3% | 44.0% | 1.33% | 1.33% | 26.4% |
| VideoAgent-Online(Iter3) | 48.0% | 21.3% | 42.0% | 1.33% | 1.33% | 26.5% |

Table 7: **Meta-World Result.** The mean success rates of VideoAgent combined with Online and Replan modules as compared to the AVDC baseline

# E ARCHITECTURAL DETAILS OF VIDEOAGENT

## E.1 VIDEO DIFFUSION TRAINING DETAILS

We use the same video diffusion architecture as the AVDC baseline. For all models, we use dropout=0, num head channels=32, train/inference timesteps=100, training objective=predict v, beta schedule=cosine, loss function=l2, min snr gamma=5, learning rate=1e-4, ema update steps=10, ema decay=0.999.

## E.2 INFERENCE TIME SPEED

In our current setup, during inference, our video generation model produces a new video within 10 seconds on a single A6000 GPU at a resolution of $128 \times 128$ for metaworld. The process of mapping this generated video to an action takes, on average, an additional 25 seconds. This action-mapping stage involves calculating optical flow, receiving feedback from the vision-language model (VLM), and to convert the video into an action sequence based on the computed flow.

| | door-open | door-close | basketball | shelf-place | btn-press | btn-press-top |
|---|---|---|---|---|---|---|
| AVDC | 30.7% | 28.0% | 21.3% | 8.00% | 34.7% | 17.3% |
| VideoAgent | 46.7% | 29.3% | 13.3% | 9.3% | 38.7% | 18.7% |
| VideoAgent-Binary | 46.7% | 32.0% | 14.7% | 6.7% | 38.7% | 21.3% |
| VideoAgent-Suggestive | 46.7% | 33.3% | 18.7% | 12.0% | 41.3% | 22.7% |
| VideoAgent-Online-Suggestive | 52.0% | 28.0% | 21.3% | 16.0% | 46.7% | 22.7% |

| | faucet-close | faucet-open | handle-press | hammer | assembly | **Overall** |
|---|---|---|---|---|---|---|
| AVDC | 12.0% | 17.3% | 41.3% | 0.00% | 5.30% | 19.6% |
| VideoAgent | 46.7% | 12.0% | 36.0% | 0.00% | 1.3% | 22.9% |
| VideoAgent-Binary | 46.7% | 17.3% | 32% | 0.00% | 5.3% | 23.8% |
| VideoAgent-Suggestive | 48.7% | 17.3% | 46.7% | 0.00% | 5.3% | 26.6% |
| VideoAgent-Online-Suggestive | 45.3% | 20.0% | 48.0% | 2.7% | 5.3% | 27.4% |

Table 8: **Meta-World: VideoAgent-Feedback Guided Results** The mean success rates for various tasks, comparing different VideoAgent-Feedback Guided variants and the AVDC baseline.

# F  VLM Feedback for Correction

# G  Details of Human Evaluation on BridgeData V2

**Qualitative Evaluation.** Next, we qualitatively evaluate video generation quality using the five Video-Score dimensions: Visual Quality (VQ) for clarity and resolution, Temporal Consistency (TC) for smooth frame transitions, Dynamic Degree (DD) for capturing accurate object/environment changes, Text-to-Video Alignment (TVA) for matching the video to the prompt, and Factual Consistency (FC) for adherence to physical laws and real-world facts. Videos are rated on a 4-point scale based on the metric in He et al. (2024): 1 (Bad), 2 (Average), 3 (Good), and 4 (Perfect). Our evaluation is based on 50 generated videos from a held-out set.

Table 9: Task Success and Other Fine-grained Human Evaluation Metrics on BridgeData-V2

| **Metrics** | | **AVDC** | **Video Agent** |
|---|---|---|---|
| **Task Success via Human Eval** | | 42.0% | **64.0%** |
| **Holistic Assessment via Human Eval** | Visual Quality | 1.74 | 1.84 |
| | Temporal Consistency | 1.58 | 1.76 |
| | Dynamic Degree | 3.14 | 2.98 |
| | Text to Video Alignment | 2.66 | 3.04 |
| | Factual Consistency | 3.22 | 3.30 |
| | **Human Eval Average** | 2.47 | **2.98** |

In terms of VQ and TC, both the baseline AVDC and our VideoAgent generate average quality videos (graded 2), with AVDC hallucinating more and generating some choppy jumps in videos temporally (we grade such videos as 1) and Video Agent fixing some of these upon video conditioned iterative refinement. The reason for AVDC baseline having higher DD is attributed to unruly movements that cause higher DD scores compared to VideoAgent, where movements are smoother. This also explains the result in fifth row of Table 5, and upon closer examination of the generated videos and their corresponding individual scores, we observed similar traits in videos having higher DD due to unnatural robot arm movements and object impermanence. TVA shows trends similar to ClipScore in Table 5 due to the better instruction following ability of VideoAgent leading to more controlled generation. FC is a very crucial metric for deployment of video generation agents as policy for task completion in robotics, scene navigation, and so on. Improved visual quality does not imply adherence to correct physical laws and real-world constraints, FC particularly checks for this aspect and due to video conditioned self-refinement, VideoAgent has better FC compared to AVDC.

## H EXAMPLES

### H.1 ZERO-SHOT GENERALIZATION ON REAL-WORLD SCENES

VideoAgent trained on Bridge dataset demonstrates strong performance on zero shot video generation for natural distribution shifts and longer language instructions. Some examples of the synthesized videos can be found in Fig. 8.

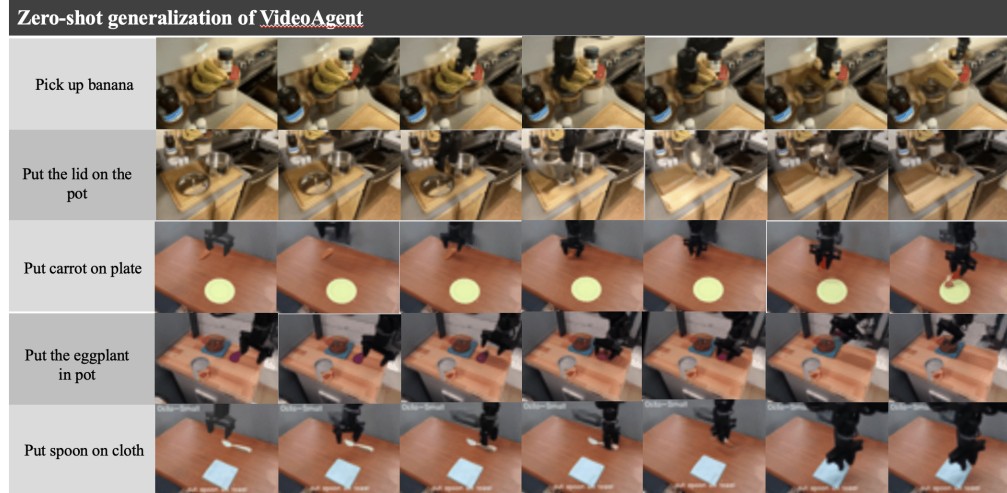

Figure 8: **Zero-shot generalization of VideoAgent:** VideoAgent generalizes fairly well to natural distribution shifts and is able to generate successful trajectories on data it has not been trained on.

### H.2 IMPROVEMENTS IN META-WORLD

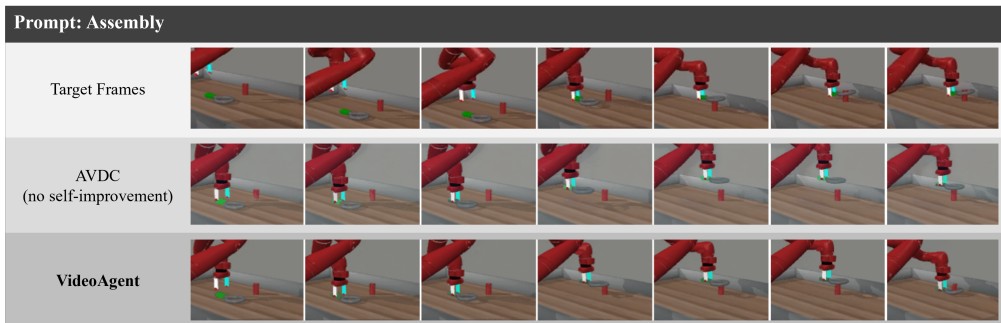

Figure 9: **Correcting Hallucinations in Video Generation:** The goal prompt is "Assembly" as shown in the Target Video. The AVDC model has problem of object permanence and action incomplete in last frame. In contrast, our VideoAgent model accurately object permanence and correctly places the inside the peg properly.

### H.3 IMPROVEMENTS IN ITHOR

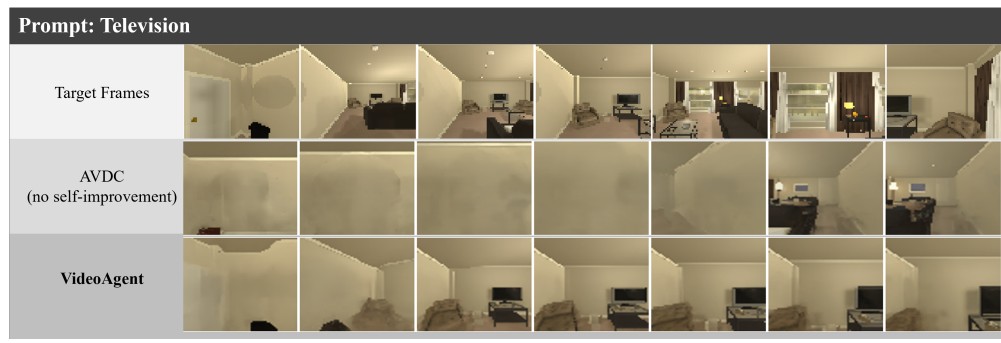

Figure 10: **Correcting Hallucinations in Video Generation:** The goal prompt is "Television" as shown in the Target Video, the goal is for the navigator to locate the object and reach near it. The AVDC model has difficulty reconstructing and navigating in the livingroom to find the television. In contrast, our VideoAgent model solves the initial frame hallucinations and accurately reaches near the television correctly.

## H.4 IDENTIFICATION AND SUGGESTIVE FEEDBACK EXAMPLES

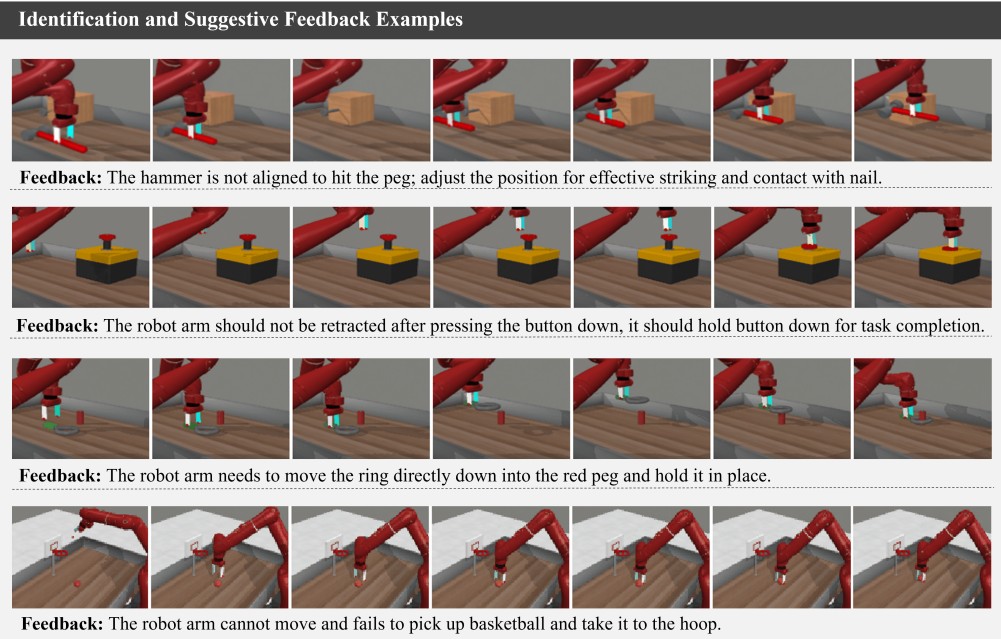

Figure 11: **Detailed VLM Feedback:** We show the efficacy of VLMs to provide useful feedback even in the absence of access to a simulator or real-world execution environment. The VLM acts as a proxy reward model to condition VideoAgent on useful corrective signals, leading to improved performance as described in Table 3.

