# OpenReview forum: "VideoAgent: Self-Improving Video Generation"
_ICLR.cc/2025/Conference — Submitted to ICLR 2025_

### Official Review · Reviewer_AFbw · 2024-10-26

**Soundness:** 3
**Presentation:** 4
**Contribution:** 3
**Rating:** 3
**Confidence:** 4

**Summary:**

This paper proposes VideoAgent for self-improving generated video plans based on external feedback. It first refines the generated video plans using self-conditioning consistency and utilizes feedback from a pretrained vision-language model (VLM), then it collects additional data from the environment to further improve video plan generation. From both simulation and real-world video generation experiments, it shows the effectiveness of the proposed method.

**Strengths:**

1. Clear equation writing, which shows the strong technique capabilities of the authors.

2. Figure 1 is clear, which makes me understand the main idea of this paper quickly.

3. Experiments for the VLM feedback is interesting.

**Weaknesses:**

1. I am unsure whether the *self-conditioning consistency* is an original contribution of this paper because of the poor writing clarity of Section 3.1. From the author's writing, it seems like some other papers also endow similar ideas. If the authors want to claim this as part of their contribution, the author should improve the writing of this part to clearly show which idea or equation is original.

2. I am confused about how to use the feedback from VLM for Equation 10. The authors did not provide any methodological description in Section 3.1 or Appendix, e.g., what is the *feedback*, or how to put the *feedback* into Equation 9.

3. Also, I am unsure whether using VLM feedback to improve video generation models is original since the author didn't discuss any related works for this part. If this is novel (or at least part of this is novel), the author should clearly point out which part is novel. Currently, people may assume that this section is totally based on some existing methods, because the author seems to assume that readers know how Equation 10 works, and therefore does not provide any method explanation.

4. From my point of view, the main contribution (if the above 3 questions can be solved) are: 1) self-consistency conditioning; 2) VLM-feedback; 3) real-world interaction feedback. However, from Table 1, we can see these three modules didn't bring much improvement (VideoAgent v.s. AVDC, only 2.7% increase). Instead, the main improvement comes from Online and Replan (23.5% increase for AVDC and 27.7% for VideoAgent). What's more, Online-and-Replan works across two different video generation methods (AVDC and VideoAgent). Thus, this greatly reduces the value of the three main contributions of this article.

**Questions:**

See the weakness part.

---

> ### Author Response · Authors · 2024-11-23
>
> Thank you for the detailed review. Please see our response below.
>
> # W1. Originality of self-conditioning consistency
> Thank you for highlighting this point. Self-conditioning consistency for videos is indeed an original contribution of this work, designed to enable iterative refinement of generated videos. In VideoAgent, we use "consistency" in a specific context to mean mapping any hallucinated or incoherent video to a realistic, non-hallucinatory one. We do not employ traditional consistency models, our iterative refinement process serves a similar purpose: it aligns multiple flawed versions of a generated video into a progressively more coherent final output. We understand that the current writing may have created some ambiguity about its novelty.
> In particular, equation 6 adapts the DDIM equation for generating $\mathbf{x}^{(t−1​)}$ from $\mathbf{x}^{(t)}$​. In our approach, we replace the "predicted $\mathbf{x}^{(0)}$" term in the original DDIM equation with $\mathbf{\hat{x}}$, which is the previously generated video sample. During training, DDIM predicts $\mathbf{x}^{(t−1)}$​ based on this $\mathbf{x}^{(0)}$, while in VideoAgent, we predict the next video based on $\mathbf{\hat{x}}$ for a set of parameters $\theta$. This adaptation is what we mean by "mimicking ODE solvers," as our method similarly refines video predictions iteratively based on previous samples.
>
> We recognize that this intuition may not be fully clear in the current text. To address this, we have revised Section 3.1 to explicitly highlight this chain of ideas and clearly delineate our original contributions.
>
> # W2. Use of VLM feedback in Equation 10
> Thank you for the suggestion. After reviewing your feedback, we have updated the paper to explicitly incorporate how feedback is conditioned into Algorithm 2. Specifically, as outlined in Equation 10 and Section 3.1, the feedback is provided in two forms: (1) binary feedback from the VLM indicating success or failure, and (2) descriptive feedback where the VLM identifies inconsistencies and errors in the generated video plan. This textual feedback is concatenated with the task condition and used to refine the video plan generation. We appreciate your suggestion, which has helped improve the clarity of our work.
>
> # W3. Novelty of using VLM feedback for video generation
> Thank you for your comment. To the best of our knowledge,the use of VLM feedback to improve video generation models is indeed a novel aspect of our work. We have updated the paper to explicitly highlight this novelty, along with the introduction of self-conditioning consistency, as the core contributions of our approach. Section 4.3.1 presents the results validating the effectiveness of these contributions.
>
> # W4. Contributions of key modules versus Online-and-Replan
> Thank you for your valuable feedback. One of the key contributions of our work is the use of **robotic tasks to ground video generation**, providing a practical framework for refining video plans in a task-specific manner. While the improvements from Online-and-Replan are significant, our refinement modules play a crucial role in offline contexts, where replanning is not feasible. This enables iterative enhancements to video plans without requiring task resets, making the approach broadly applicable across various embodiments and data inputs. Additionally, our model demonstrates strong **zero-shot generalization** on real-world tasks (shown in Appendix H1 of the updated version of our paper), further highlighting its robustness and versatility in diverse settings.
>
> a) **Online Environment Access:** Unlike traditional reinforcement learning, which relies on exploration through trial and error, our method focuses on goal-conditioned video generation to enhance agent performance. By leveraging real-world feedback, we refine video plans to align closely with intended goals, minimizing the need for computationally expensive trial and error. This approach turns online interaction into a direct optimization of video fidelity and task success, offering a more efficient use of environment access. Moreover this approach is scalable and adaptable to various robot embodiments, and potentially even internet scale human data.
>
> b) **Architecture and Objectives:** Our architecture addresses a critical challenge: as video generation models improve, resolving inaccuracies within video plans will become the primary bottleneck. We proactively address this with self-conditioning consistency and VLM feedback, refining video quality iteratively and enabling high-precision plan generation. This forward-looking approach positions our method to adapt as video generation technology advances, providing impactful, scalable improvements in video-agent learning.

---

> > ### Comment · Reviewer_AFbw · 2024-11-25
> >
> > Thanks for addressing my questions regarding the novelty and technical details.
> >
> > However, I do not think your W4 is reasonable. This is related to the main contribution, thus I maintain my score. My questions are:
> >
> > 1. The authors said the main contribution of this paper is *the use of robotic tasks to ground video generation* rather than improving the success rates of robot manipulation tasks using the generated high-quality videos. This is weird because most of the experiments in this paper are showing the success rates (Table 1, 6, 7, 8).
> >
> > 2. If this is a video generation paper, the authors should report some results of the generated video quality such as SSIM, PSNR, and FVD. Currently, the generated videos are only qualitatively evaluated by VLM and humans, which is not convincing because nobody can reproduce these results.
> >
> > 3. The authors claim in a) that the advantage of this paper is *minimizing the need for computationally expensive trial and error* and *offering a more efficient use of environment access*. However, there are no experiments showing the sample efficiency of the proposed method. Thus this point is unverified and can just be called a hypothesis.
> >
> > Besides these questions, I want to know if there are other works that also aim to generate accurate videos. If true, the authors should use them as baselines. Otherwise the authors should clearly state in the paper that they are the first paper to address this problem.

---

> > > ### Author Response · Authors · 2024-12-01
> > >
> > > ## Video grounding vs. Task success rates
> > > Thank you for raising this question! Using robotics to ground video generation is the key contribution of this paper, but it is not the only one. Our work introduces novel mechanisms, such as self-conditioning consistency and feedback-guided refinement, which enhance video generation in task-specific contexts. Also to the best of our knowledge, we are the first to use VLM feedback to improve video generation and also guide the test-time self-refinement process. These contributions extend beyond robotics, but grounding video generation through robotic tasks allows us to demonstrate the practical utility of the generated videos in real-world scenarios. The results show that our method leads to measurable improvements in task success rates, underscoring its effectiveness.
> > >
> > > ## Quantitative evaluation metrics for generated videos
> > > This paper is fundamentally about video generation for robotics, so we focused on reporting scores relevant to this domain. Alongside task success rates, we included metrics such as CLIP-based evaluations and VideoScore[1] for the Bridge dataset to assess semantic alignment, factual consistency, coherence, and overall video quality. These metrics better reflect the requirements of task-specific video plans in robotic applications.
> > >
> > > ## Sample efficiency and environment access
> > > We agree that our claim about minimizing computationally expensive trial-and-error and improving sample efficiency is currently a hypothesis. While our iterative refinement process experimentally reduces environment interactions by enabling offline improvements, we acknowledge the need for more explicit validation. We plan to include such experiments in future work to substantiate this claim.
> > >
> > > Besides VideoAgent, there are other works that use video generation as policy[2,3,4,5,6]; however, generated videos often hallucinate or violate physical laws. None of these works focuses on improving video generation quality itself. We identified this as a limiting factor in advancing research in this domain and addressed it by prioritizing video generation refinement as a core component of our approach. We thank you for raising these points, and we hope the advancements presented in this work, combined with our outlined future directions, provide a compelling case for the value of this research. We hope this will result in a further favourable assessment of our work. Thanks again!
> > >
> > > [1] He, Xuan, et al. "Videoscore: Building automatic metrics to simulate fine-grained human feedback for video generation." arXiv preprint arXiv:2406.15252 (2024).
> > >
> > > [2] Yang, Sherry, et al. "Video as the New Language for Real-World Decision Making." arXiv preprint arXiv:2402.17139 (2024).
> > >
> > > [3] Du, Yilun, et al. "Video language planning." arXiv preprint arXiv:2310.10625 (2023).
> > >
> > > [4] Luo, Yunhao, and Yilun Du. "Grounding Video Models to Actions through Goal Conditioned Exploration." arXiv preprint arXiv:2411.07223 (2024).
> > >
> > > [5] Ko, Po-Chen, et al. "Learning to act from actionless videos through dense correspondences." arXiv preprint arXiv:2310.08576 (2023).
> > >
> > > [6] Du, Yilun, et al. "Learning universal policies via text-guided video generation." Advances in Neural Information Processing Systems 36 (2024).

---

> > > > ### Comment · Reviewer_AFbw · 2024-12-03
> > > >
> > > > Thanks for the authors' response. My concerns have not been solved by your response. Since the rebuttal period is going to end, here I summarize my main concern for ACs and PCs when they make the decision:
> > > >
> > > > This paper proposes to use robotics to ground video generation, which includes: 1) self-consistency conditioning; 2) VLM feedback; 3) real-world interaction feedback. However, current experiments cannot convince me that these methods are effective, either for robotics tasks or for improving the video generation quality, because:
> > > >
> > > > 1. The success rates results in Table 1 show that these three modules didn't bring much improvement (VideoAgent v.s. AVDC, only 2.7% increase). Instead, the main improvement comes from Online and Replan (23.5% increase for AVDC and 27.7% for VideoAgent). What's more, Online-and-Replan works across two different video generation methods (AVDC and VideoAgent).
> > > >
> > > > 2. No quantitative video generation quality experiment results (such as PSNR, SSIM, and FVD metrics). Only qualitative results evaluated by humans and VLMs are reported, which are difficult for the community to reproduce.
> > > >
> > > > Thus, I think although the proposed approach is interesting, there is a lack of experimental evidence to support its effectiveness.

---

### Official Review · Reviewer_jLDg · 2024-11-03

**Soundness:** 3
**Presentation:** 2
**Contribution:** 3
**Rating:** 5
**Confidence:** 4

**Summary:**

The paper introduces VideoAgent which self-improves generated video plans based on external feedback.
Given an initial frame and a language instruction, VideoAgent first generates a video plan and then iteratively refines it based on feedback from a pre-trained VLM.
The refined video plan can be converted to low-level robot actions for execution.
Successful trajectories are added to the dataset for further fine-tuning the video generation and refinement models.
Experiments on video plan generation were performed in both simulation and real-robot videos.
Results show that the proposed method can reduce hallucination in video plan generation and thus boost performance in downstream manipulation tasks.

**Strengths:**

The paper presents a novel method which uses feedback from VLMs to refine generated video plans.
It introduces a self-improving consistency model which predicts the clean video from a generated video and feedback from the VLM.
The proposed method can be continuously improved through online fine-tuning.
Experiment results indicate that the proposed method effectively enhances video generation and improves performance on downstream manipulation tasks.
The paper also includes ablation studies to examine the impact of different types of feedback from the VLM, and the effect of refinement and online iterations.
Additionally, it offers an in-depth analysis of the VLM's performance in providing feedback for video generation.

**Weaknesses:**

1. The paper leverages a self-conditioning consistency loss (Eqn. 7) for video refinement. The first term in Eqn. 7 is a diffusion loss and the second term is for consistency. The reason why including the second term in Eqn. 7 is not very clear. It seems that it encourages generating consistent $x^{(0)}$ from different $\hat x_{i}$. Is it possible to provide more explanation on why the consistency loss is necessary to be included? It would be great to include an ablation study which compares the refinement performance with and without this consistency loss.

2. Figure 2 is not very clear. If I understood correctly, $x^{(t + \Delta)}, x^{(t)}, x^{(t - \Delta)}$ are noisy latents at different denoising timesteps in a diffusion. But what is the target of this diffusion process? Is it $\hat x_{(i+1)}$? It would be great to provide more descriptions in the caption for further clarification.

3. The paper lacks real-robot experiments to validate the effectiveness of the proposed method in real-world policy learning. The paper would benefit strongly from incorporating a real-robot policy learning experiment. And comparing the proposed method with baseline methods without self-improving consistency (e.g. AVDC) would help understand how the proposed video refinement process helps downstream policy learning in the real world.

**Questions:**

1. VLMs are prone to hallucinations in some cases. Were any hallucinations from VLM feedback observed in the experiments? If such hallucinations were observed, would they potentially mislead the refinement process?

2. How does the proposed method perform in generalization settings? Does introducing feedback from VLM help the proposed method on handling novel text instructions? It would be great to incorporate a generalization experiment on video plan generation on real-robot data (e.g. BridgeData V2).

3. In Algorithm 2, if I understood correctly, the $\pi_{\theta}$ in line 284 should be $\hat f_{\theta}$?

---

> ### Author Response · Authors · 2024-11-23
>
> Thank you for recognizing the significance of this work! Please see our response to your questions below.
> # W1. Inclusion of the self-conditioning consistency loss
> Thank you for your question. The second term in equation 7, the consistency loss, is crucial for ensuring that the generated $\mathbf{x}^{(0)}$ remains consistent when conditioned on different $\mathbf{x}_{i}$. This loss encourages the refinement process to align outputs across different intermediate inputs, improving the coherence and quality of the refined video.
>
> Including this term allows VideoAgent to handle variations in the input video plans during refinement, ensuring that the model does not diverge when faced with slightly different initial conditions. Without this consistency term, the refinement process may fail to converge or produce results with greater inconsistencies.
>
> # W2. Figure 2 and the target of the diffusion process
> Thank you for your feedback. After reviewing the original figure, we recognized that it did not fully capture the denoising and refinement processes accurately, which could cause confusion. To address this, we have updated the figure to better represent the methodology. The updated figure explicitly separates the denoising process and the refinement iterations. The denoising process progresses from $x^{(T)}$ to $x^{(t)}$, eventually generating $\mathbf{x}_i$.
>
> This represents the diffusion model's role in producing an initial video plan. In the refinement process, shown below, $\mathbf{x}_{i+1}$ is generated by conditioning on $\mathbf{x}_i$, effectively refining the video output iteratively. This iterative process ensures that each subsequent video sample incorporates corrections, guided by feedback, to achieve improved quality and coherence. This updated figure more accurately reflects the methodology described in the paper, and we have included it in the revised version for improved clarity.
>
> # W3. Absence of real-robot experiments
> Thank you for this insightful suggestion. We acknowledge that our work does not include real-robot experiments due to a lack of access to real-robot systems, as noted in the limitations section of the paper. However, our primary focus is to establish a stepping stone for improved video generation models, with the goal of facilitating future transfer to real-world robotic policy learning.
>
> The proposed video refinement process is designed to address challenges in video plan generation and self-improvement, which are critical for real-robot applications. By demonstrating significant improvements in video generation and refinement, our method lays the groundwork for future research to integrate these advancements into real-robot systems. This work aims to bridge the gap between video generation and real-world policy learning, making future real-robot deployment more feasible and effective.

---

> > ### Author Response · Authors · 2024-11-23
> >
> > # Q1. Potential hallucinations in VLM feedback
> > We thank the reviewer for raising this observation. Instances of hallucination in VLM feedback were indeed observed, particularly in experiments involving descriptive feedback. For example, when the gripper and the pot overlapped, the VLM provided feedback such as: "Adjust the gripper's path to avoid collision with the pot, ensuring a smooth transfer of the green object to the stove's right side." In this case, the VLM misunderstood object overlap as a collision, indicating an error in interpreting the generation.
> >
> > Despite this, our empirical results demonstrate consistent improvements across all experiments, indicating that the overall refinement process is robust to such inaccuracies.
> > Our iterative refinement approach plays a key role in mitigating the impact of VLM hallucinations. Even if one iteration of feedback contains inaccuracies, subsequent refinements are likely to correct and improve the video plans. The iterative nature ensures that the quality of video plans continues to improve over time. Additionally, it is improbable that the VLM would produce hallucinatory feedback consistently across all iterations, especially as the generated video plans improve with refinement.
> >
> > Our experimental design accounts for these limitations and demonstrates its effectiveness even under current VLM capabilities. We expect our method to scale as the quality of VLMs improves.
> >
> > # Q2. Generalization and the Role of VLM Feedback
> > Thank you for your question. We have conducted generalization experiments specifically on BridgeData V2, and our method demonstrates strong performance, generalizing well to both **natural distribution shifts** and **longer language instructions**. The integration of feedback from the VLM has proven particularly effective in handling novel text instructions, enabling task-specific corrections and improving video plan coherence. These results highlight the robustness of our approach, and we have included the relevant findings and discussion in Appendix H1 in the revised manuscript to address this point.
> >
> > # Q3. Clarifying Notation in Algorithm 2
> > Thank you for pointing this out. To clarify, $\pi_\theta$ in line 284 is a combination of $f_\theta$ and $\hat{f}_{\theta}$.
> >
> > It represents the overall function that integrates both components, where $f_\theta$ is responsible for video generation, and $\hat{f}_{\theta}$ handles the refinement process.

---

> > > ### Comment · Reviewer_jLDg · 2024-11-25
> > > **Response to the Authors**
> > >
> > > Thank you for addressing my comments and questions. Most of my concerns are addressed. The updated figure 2 is clear and figure 8 showcases the zero-shot generalization capabilities of the proposed method in real-world scenes. However, I believe a systematic real-robot experiment is necessary to validate the effectiveness of the proposed method in policy learning. Overall, I maintain my rating.

---

> > > > ### Author Response · Authors · 2024-12-01
> > > >
> > > > ## Future directions
> > > > We agree that systematic real-robot experiments are an important next step to further validate the effectiveness of our method in policy learning. While we were unable to include such experiments in the current study due to hardware access constraints, we have aimed to demonstrate the broad applicability of VideoAgent through experiments in MetaWorld, iThor, and real-world datasets. These results highlight the potential of VideoAgent to bridge the gap between video generation and robotic control.
> > > >
> > > > We value your feedback and will use it to guide our future work, where we aim to incorporate real-robot experiments and further enhance the practical applicability of our method. We hope this will result in a further favourable assessment of our work. Thanks again!

---

### Official Review · Reviewer_b49X · 2024-11-03

**Soundness:** 2
**Presentation:** 2
**Contribution:** 2
**Rating:** 5
**Confidence:** 4

**Summary:**

This paper proposes VideoAgent, a model designed to improve video generation quality based on feedback from a VLM. Specifically, VideoAgent trains both a video generation model and a video refinement model, which share parameters and are based on diffusion processes. The video generation model functions as a conventional text-to-video model, while the video refinement model takes as input the ground-truth video, the predicted video, and feedback from the VLM to produce a refined video. The final output video can be mapped to an action trajectory. When the environment is interactive and can confirm task success, VideoAgent performs rollouts within the environment to gather additional data for training (but very slow now). This method is evaluated at the policy level in the Meta-World and iTHOR benchmarks, and for video quality on the Bridge dataset.

**Strengths:**

1. The idea of VideoAgent is novel; I haven’t seen an iterative approach to improving video generation quality before.
2. Obtaining feedback on videos from VLM is feasible.

**Weaknesses:**

1. The writing section can be strengthened. The distinction between the consistency model and DDPM is vague in this paper, yet it is crucial. Given that I believe DDPM + DDIM can achieve the same results (referring to the implementation of video generation models and video refinement models), it’s even more necessary to explain the necessity and motivation for using the consistency model. Specific issues can be referenced in the questions section.

2. The experimental section of the paper is relatively weak. The baseline is only AVDC, a typical video generation model without a corresponding video refinement model. Possible baselines could include VLP[1] and a naive text-to-video diffusion model (under similar training and inference FLOP). Additionally, results from traditional baselines based on BC and RL should also be included for reference.

3. More realistic robotic manipulation benchmarks should be considered, such as Simper[2].

4. It would be better to conduct experiments on a real robot; currently, the focus is only on testing quality using real-robot videos. Success rate evaluations are missing, which may necessitate the use of a real robot. If a real robot is not available, providing results from Simper would be acceptable.

5. Even after refinement, the video generation quality of VideoAgent on the Bridge dataset is not high compared to modern classic diffusion models, and the resolution is very low.

6. Fundamentally, generating a video and then generating actions is not widely regarded as a correct approach; the improvements made in this paper are still limited by the fundamental flaws of video planning. The video generation is too slow to be suitable as a policy. If the paper could demonstrate the importance of video generation for policy, it would greatly strengthen its contributions.

7. The paper seems to avoid the discussion of video generation speed. In reality, how many actions can be generated per second with this method? Or what is the average time required to generate a single action? The resolution should also be reported.

[1] Video Language Planning.
[2] Evaluating Real-World Robot Manipulation Policies in Simulation

**Questions:**

When I read this paper, I encountered some areas of confusion, and I hope my feedback can help the authors improve the clarity of the writing. If there are any inaccuracies in my understanding, I welcome any corrections.

1. The motivation for introducing the consistency model should be clearly articulated. I didn’t see a compelling reason to use the consistency model, as it seems possible to achieve similar results by using the training objective of DDPM and employing DDIM for inference.

2. Lines 119-120 should describe the loss for the consistency model rather than for DDPM. In DDPM, the loss should be calculated from ($x^{t+1}$ to $x^t$), which also makes the "vanilla objective for video diffusion" mentioned in line 197 confusing.

3. In line 224, it seems that video generation should be a single step if using a consistency model. Why are multiple steps required, and if they are, what distinguishes this approach from DDIM?

4. The motivation in Section 3.1 could be more clearly defined. The overall objective of the proposed method is to (1) generate video and (2) refine the video based on feedback until it is accepted by the VLM, using the same parameters. However, this goal is not clearly explained at the beginning of Section 3.1, which requires readers to infer the motivation after understanding the method.

5. Line 164 states, "the model can learn to preserve the realistic part of the video while refining the hallucinatory part," which could be misleading. Feedback is applied to the entire video, not frame-level feedback. This statement may somewhat overstate the model’s capability.

6. It is unclear how the parameters are shared between the video generation model and the video refinement model.

7. It is not shown in the method how the video maps to action. Indicating where this is discussed in the paper would be helpful.

---

> ### Author Response · Authors · 2024-11-23
>
> Thank you for the detailed review. Please see our response below.
> # W1. Consistency model vs. DDPM
> In VideoAgent, we use *"consistency"* in a specific context to mean mapping any hallucinated or incoherent video to a realistic, non-hallucinatory one. Although we do not employ traditional consistency models, our iterative refinement process serves a similar purpose: it aligns multiple flawed versions of a generated video into a progressively more coherent final output.
>
> Each iteration in VideoAgent can be viewed as an analogue to a consistency model timestep. Where traditional consistency models aim to map noisy representations at any given timestep to a clean data point, our approach uses each iteration to refine and align previously generated, flawed video frames toward a coherent, realistic final video.
>
> While we adopt the **DDPM + DDIM structure** during training and inference, our unique refinement goal and use of consistency distinguish VideoAgent from these frameworks. Traditional DDPM and DDIM models do not perform iterative refinement—if a sampled video is flawed, the output remains flawed, as these models lack mechanisms to self-correct.
>
> In contrast, the consistency mechanism in VideoAgent, captured in the second part of **equation 7**, applies a loss function to minimize differences between videos generated by conditioning on two individual samples at different timesteps. This iterative process promotes frame-to-frame coherence and allows VideoAgent to refine outputs progressively, addressing errors and improving video quality—a capability that DDPM and DDIM structures alone cannot achieve.
>
> We recognize that this intuition may not be fully clear in the current text. To address this, we will revise Section 3.1 to explicitly highlight this chain of ideas and clearly delineate our original contributions.
>
> # W2. Addressing the need for comprehensive baseline comparisons
> Thank you for your valuable feedback. After reading your comments, we conducted additional experiments on the suggested baselines, including VLP and Behavioral Cloning (BC). AVDC is already state-of-the-art compared to BC (16.2%) and UniPi (6.1%). However, it falls short of the overall success rate achieved by Diffusion Policy (24.1%) without replanning. In contrast, VideoAgent surpasses Diffusion Policy during its online iterations, even without employing replanning. This demonstrates the effectiveness of our iterative refinement approach in improving video plan quality and task success, highlighting the robustness of VideoAgent in comparison to existing methods. The results from these experiments have been included in the updated manuscript and have been uploaded alongside this response for your reference.
>
> We would like to highlight key distinctions between our approach and VLP. While VLP uses language for planning, our method leverages language as feedback to iteratively refine video plans—a fundamental difference in methodology. Additionally, VLP does not perform refinement, which is central to our approach. Despite these, we include VLP results for comparison. To ensure fairness, we also present results from a modified version of AVDC integrated with a VLM to select the best video plan without using refinement iterations.
>
> ## Meta-World Results
> | Method                   | door-open | door-close | basketball | shelf-place | btn-press | btn-press-top | faucet-close | faucet-open | handle-press | hammer | assembly | **Overall** |
> |--------------------------|-----------|------------|------------|-------------|-----------|---------------|--------------|-------------|--------------|--------|----------|-------------|
> | BC-Scratch              | 21.3%    | 36.0%     | 0.0%       | 0.0%        | 34.7%    | 12.0%        | 18.7%       | 22.7%      | 28.0%       | 0.0%   | 0.0%     | 15.4%      |
> | BC-R3M                  | 1.3%     | 58.7%     | 0.0%       | 0.0%        | 36.0%    | 4.0%         | 18.7%       | 17.3%      | 37.3%       | 0.0%   | 1.3%     | 16.2%      |
> | UniPi (with Replan)     | 0.0%     | 36.0%     | 0.0%       | 0.0%        | 6.7%     | 0.0%         | 4.0%        | 9.3%       | 13.3%       | 4.0%   | 0.0%     | 6.1%       |
> | VLP                     | 33.3%    | 28.0%     | 17.3%      | 8.0%        | 36.0%    | 18.7%        | 30.7%       | 10.7%      | 33.3%       | 0.0%   | 1.3%     | 19.8%      |
> | Diffusion Policy                     | 45.3%    | 45.3%     | 8.0%      | 0.0%        | 40.0%    | 18.7%        | 22.7%       | 58.7%      | 21.3%       | 4.0%   | 1.3%     | 24.1%      |

---

> > ### Author Response · Authors · 2024-11-23
> >
> > # W3 and W4. Real world robot manipulation experiments using Simpler
> > Thank you for your feedback regarding testing on more realistic robotic setups. Due to the unavailability of real robots and time constraints, we focused on zero-shot video generation in SimplerEnv, demonstrating that VideoAgent can generate realistic and plausible video plans for tasks within this environment. These results highlight the model’s generalizability to realistic robotic environments. Specific examples of these video plans have been included in Appendix H1 for reference.
> >
> > While we could not execute actions in SimplerEnv within the current time frame, we plan to train an inverse dynamics model in the future to extract actions from VideoAgent-generated video plans and validate task execution in SimplerEnv. This will allow us to provide a more comprehensive evaluation of our approach, including success rate assessments.
> >
> > # W5. Video generation quality and resolution limitations
> > The goal of this work is to introduce the concept of video refinement. We perform our analysis based on the baseline architecture of AVDC and show improvements on that. To make fair comparisons with the baseline, we keep the resolution exactly the same as what they have trained their model on. Training the best video generation model is not the objective of this work.
> >
> > # W6. Role of video generation in policy learning
> > Thank you for your feedback. While video generation for policy learning might not have been a traditional approach in the past, recent advancements highlight its growing relevance and utility. Several recent works have successfully employed video generation as a foundation for decision-making, demonstrating its ability to model complex, high-dimensional environments. These include approaches that leverage video generation to predict future states, understand task dynamics, and enable grounded reasoning for policy learning [1,2,3,4,5,6,7,8, 9, 10,11].
> >
> > We have included additional references and discussion on this topic in the Related Works section to further support the significance of video generation for policy learning.
> >
> > [1] Yang, Sherry, et al. "Video as the New Language for Real-World Decision Making." arXiv preprint arXiv:2402.17139 (2024).
> > [2] Du, Yilun, et al. "Video language planning." arXiv preprint arXiv:2310.10625 (2023).
> > [3] Luo, Yunhao, and Yilun Du. "Grounding Video Models to Actions through Goal Conditioned Exploration." arXiv preprint arXiv:2411.07223 (2024).
> > [4] Ko, Po-Chen, et al. "Learning to act from actionless videos through dense correspondences." arXiv preprint arXiv:2310.08576 (2023).
> > [5] Du, Yilun, et al. "Learning universal policies via text-guided video generation." Advances in Neural Information Processing Systems 36 (2024).
> > [6] Sun, Shao-Hua, et al. "Neural program synthesis from diverse demonstration videos." International Conference on Machine Learning. PMLR, 2018.
> > [7] Du, Yilun, et al. "Learning universal policies via text-guided video generation." Advances in Neural Information Processing Systems 36 (2024).
> > [8] Nair, Suraj, et al. "R3m: A universal visual representation for robot manipulation." arXiv preprint arXiv:2203.12601 (2022).
> > [9] Bahl, S., Gupta, A., & Pathak, D. (2022). Human-to-robot imitation in the wild. arXiv preprint arXiv:2207.09450.
> > [10] Bharadhwaj, Homanga, et al. "Gen2act: Human video generation in novel scenarios enables generalizable robot manipulation." arXiv preprint arXiv:2409.16283 (2024).
> > [11] Ye, Seonghyeon, et al. "Latent Action Pretraining From Videos." arXiv preprint arXiv:2410.11758 (2024).
> >
> > # W7. Video generation speed and resolution
> >  In our current setup, during inference, our video generation model produces a new video within 10 seconds on a single A6000 GPU at a resolution of 128x128 for metaworld. It can run on a 12GB gpu card as well. The process of mapping this generated video to an action takes, on average, an additional 25 seconds. This action-mapping stage involves calculating optical flow, receiving feedback from the vision-language model (VLM), and to convert the video into an action sequence based on the computed flow. Thank you for pointing this out. We have included this detail in the updated paper.

---

> > > ### Author Response · Authors · 2024-11-23
> > >
> > > # Q1. Motivation and role of the consistency model
> > > Thank you for raising this point. The motivation for introducing the consistency mechanism in VideoAgent aligns with the way **large language models (LLMs)** require more than just pretraining objectives to perform well. While pretraining objectives provide a strong foundation, additional techniques like **supervised fine-tuning (SFT)** and **reinforcement learning from human feedback (RLHF)** are necessary to ground the model and enable it to self-correct. Similarly, while the DDPM training objective provides a base for video generation, it is insufficient for addressing errors or hallucinations in generated samples.
> > > The consistency mechanism introduces a complementary refinement process, enabling VideoAgent to self-correct iteratively. **Generation and refinement are orthogonal—both are needed** to achieve robust and coherent outputs. While DDIM accelerates inference, it cannot refine bad samples; the addition of consistency allows VideoAgent to handle imperfections and progressively improve video plans. We will revise the paper to clearly articulate this motivation.
> > >
> > > # Q2.  Loss function and "vanilla objective" terminology
> > > As previously noted, we use the **DDPM objective** for training our model. During training, the model can learn to either predict $\mathbf{x}^{(t)}$​ given $\mathbf{x}^{(t+1)}$​ or predict $\mathbf{x}^{(0)}$​ directly given $\mathbf{x}^{(t)}$​. Our approach implements the second objective, where we formulated the loss function to predict $\mathbf{x}^{(0)}$​ directly from $\mathbf{x}^{(t)}$​.
> > >
> > > # Q3. Explaining multi-step refinement in the context of consistency models
> > > Our model can indeed generate refined videos in a single step similar to consistency models. But as mentioned in the consistency models paper itself, the quality of generation can be improved by increasing the number of timesteps. Similarly, we can generate better refined videos with more iterations.
> > >
> > > # Q4. Motivation and objectives in section 3.1
> > > We would like to clarify that Section 3.1 is dedicated to describing the methods we employed, including the loss computations and the integration of feedback as additional conditioning to improve video generation. The objective of refining videos based on VLM feedback is specifically introduced in Section 3.2, where we explain how the VLM judges the refined videos and terminates iterations once an acceptable video is generated. Line 72-73 states and the corresponding paragraph states the motivation behind VideoAgent.
> > >
> > > To enhance readability, we have added an introductory statement at the beginning of Section 3.1 to briefly outline the overall objective of our approach, including both the video generation and VLM-guided refinement. This should provide a clearer framework for understanding the motivation of each section.
> > >
> > > # Q5. Scope of feedback application and model capabilities
> > > Thank you for highlighting this point. We agree that the feedback is applied to the entire video rather than on a frame-by-frame basis. The statement "preserve the realistic part of the video while refining the hallucinatory part" provides an intuition for why refinement makes sense, though it may not precisely describe the underlying mechanism. To clarify, self-conditioning involves conditioning on the previously generated video, allowing the model to iteratively refine outputs. This process helps retain the realistic portions of the video while focusing on correcting inconsistencies in less accurate regions.
> > >
> > > Our loss function implicitly prioritizes frames with larger inconsistencies by penalizing deviations from the target frames more heavily, enabling the model to learn to align better with the desired video plan. To address this, we have updated the paper to clarify that this explanation serves as an intuitive framework rather than a strict depiction of the refinement process.
> > >
> > > # Q6. Parameter sharing between video generation and refinement models
> > > Since the video generation model and the video refinement model take the same input and output, we can indeed share their parameters. A single model architecture is capable of generating videos and refining it. This is another motivation behind using “consistency models” as they are capable of doing something similar.
> > >
> > > # Q7. Video-to-action mapping process
> > > Section 3.3 sheds some light on how the videos are mapped to action. As stated in eqn 15, we use an optical flow model which maps the videos to control level actions. Based on these predicted flows, which essentially gives us a dense prediction of pixel movements, we can reconstruct object movements and robot movements in the video. We use the GMFlow model to predict the optical flow, similar to the AVDC baseline. We have also included this information in Appendix D1 in the updated version of our paper.

---

> > > ### Comment · Reviewer_b49X · 2024-11-25
> > >
> > > **W3 and W4. Real-world robot manipulation experiments using Simpler**
> > > I reviewed the results in Appendix H1. However, I found the videos were somewhat blurry. And I am confused about the concept of "zero-shot generalization" described here. If it refers to generalizing to unseen trajectories, this might not align with the common definition of zero-shot generalization in world models[1]. Zero-shot generalization typically implies generalization to unseen objects, language instructions, or even scenes. Given that the video agent has not been pretrained, it might lack such generalization capabilities. However, this point is not a drawback.
> > >
> > > **W7. Video generation speed and resolution**
> > > How many executable actions can be generated in a single inference step? This is crucial for determining the action FPS.
> > >
> > > [1] Vista: A Generalizable Driving World Model with High Fidelity and Versatile Controllability

---

> > ### Comment · Reviewer_b49X · 2024-11-25
> >
> > **W1. Consistency Model vs. DDPM**
> > Thank you for your response; I fully understand the approach you proposed. I suspect that "Refinement Generation" might be an appropriate term. Considering that "Consistency Models" could be interpreted as [1].
> >
> > **W2. Addressing the Need for Comprehensive Baseline Comparisons**
> > Can this experiment be understood as achieving a 50% success rate for VideoAgent-Online-Replan in MetaWorld, while Diffusion Policy achieves only 24.1%? The improvement seems quite significant but is limited to MetaWorld. If the improvement were demonstrated on a widely recognized benchmark, it would be far more convincing. The resolution and dataset size of MetaWorld are relatively small.
> >
> > [1] Consistency Models, Yang Song, arXiv 2303.01469.

---

> ### Comment · Reviewer_b49X · 2024-11-25
>
> Thank you for your response. Most of my questions were based on confusion regarding the consistency model, which has now been resolved.
>
> I would like to summarize my thoughts on this paper:
>
> 1. The paper introduces a novel method to refine the generated videos through a refine model.
> 2. Experiments on metaworld demonstrate its superiority compared to AVDC (ICLR 2024) and Diffusion Policy.
>
> While the authors' improvements have enhanced the quality of the paper, I believe it lacks a comparison in real-world robotic settings, such as Calvin[1] or Simpler, or real robots. The experiments are limited to metaworld. Recent works [2][3] increasingly focus on studies based on real-world robotic setups. Compared to these works, Video-Agent is conceptually novel but may lack practical applicability, particularly since this paper is fundamentally an application-focused study. Therefore, I have decided to increase my score to 5, but I remain hesitant about accepting this paper. Due to the low action FPS, it may struggle to be applied to real robotic systems. In the future, if video generation becomes real-time, it might hold more potential.
>
> References:
> [1] CALVIN: A Benchmark for Language-Conditioned Policy Learning for Long-Horizon Robot Manipulation Tasks
>
> [2] Scaling Proprioceptive-Visual Learning with Heterogeneous Pre-trained Transformers
>
> [3] RDT-1B: A Diffusion Foundation Model for Bimanual Manipulation

---

> > ### Author Response · Authors · 2024-12-01
> >
> > ## W2. Addressing the need for comprehensive baseline comparisons
> > Thank you for emphasizing the need to evaluate on widely recognized benchmarks. To address this, we conducted experiments on the iTHOR dataset, a challenging and realistic benchmark for robotic manipulation. The results demonstrate significant improvements with VideoAgent compared to both BC and AVDC baselines. Specifically, VideoAgent achieves an overall success rate of 34.2%, outperforming AVDC (31.3%), as well as BC-Scratch (2.1%) and BC-R3M (0.4%), which struggle significantly across all room types due to their reliance on action-labeled data. We include the results below for your reference.
> > | Room           | BC-Scratch (%) | BC-R3M (%) | AVDC (%) | VideoAgent (%) |
> > |----------------|----------------|------------|----------|----------------|
> > | Kitchen        | 1.7%          | 0.0%       | 26.7%    | 28.3%          |
> > | Living Room    | 3.3%          | 0.0%       | 23.3%    | 26.7%          |
> > | Bedroom        | 1.7%          | 1.7%       | 38.3%    | 41.7%          |
> > | Bathroom       | 1.7%          | 0.0%       | 36.7%    | 40.0%          |
> > | **Overall**    | **2.1%**      | **0.4%**   | **31.3%**| **34.2%**      |
> >
> > The iTHOR results underscore the generalizability of VideoAgent to more realistic and complex benchmarks, further validating its applicability in diverse robotic scenarios. We have included these results in our updated paper.
> >
> > ## W3 and W4. Real-world robot manipulation experiments using Simpler
> > Thank you for your observations regarding the results in Appendix H1 and for raising this important question about zero-shot generalization. In the context of our work, "zero-shot generalization" refers to VideoAgent's ability to generate coherent video plans for unseen tasks or trajectories without additional finetuning on those specific tasks. While this may differ from the common definition in world models, where zero-shot typically implies generalization to entirely novel objects, instructions, or environments, our definition aligns with the task-specific context of robotic manipulation.
> >
> > We acknowledge that VideoAgent has not been pretrained on a large dataset and, as such, its zero-shot generalization capabilities are task-focused rather than encompassing broad generalization across diverse domains. Despite this, the ability of VideoAgent to generate plausible video plans for unseen tasks in toy bridge dataset used in SimplerEnv showcases its robustness and flexibility within the domain of robotic video generation. We have clarified this distinction in our revised text to ensure alignment with broader definitions and avoid potential misunderstandings.
> >
> > ## W7. Video generation speed and resolution
> > With replans, the task horizon is set to a maximum length of **500 steps**, meaning that VideoAgent is capable of generating up to 500 executable actions in a single inference.
> >
> > Thank you for your thoughtful feedback and for acknowledging the conceptual novelty of VideoAgent. We understand the importance of real-world robotic experiments and the growing focus on such setups in recent works, including CALVIN and RDT-1B. Due to hardware constraints and the logistical challenges of coordinating access to real-robot systems, we were unable to conduct real-robot experiments in the current work. However, we have taken steps to validate VideoAgent in simulated environments like MetaWorld and iThor, as well as on real-robot video datasets, demonstrating its generalizability and robustness.
> >
> > We view real-robot experiments as an important direction for future work. Our method is designed as a stepping stone to bridge video generation with robotic control, focusing on task-specific video refinement and action generation. By grounding video generation in robotic tasks, VideoAgent lays the groundwork for applications in both simulation and real-world systems.
> > Regarding the action FPS, we would like to clarify that VideoAgent performs comparably to the baseline model (AVDC) used for finetuning. Our approach generates a full trajectory of up to 500 steps for each inference, which aligns with the task requirements of simulated and real-world robotics. Therefore, we believe VideoAgent is well-suited for the current scope of robotic applications.
> >
> > We thank you for raising these points, and we hope the advancements presented in this work, combined with our outlined future directions, provide a compelling case for the value of this research. We hope this will result in a further favourable assessment of our work. Thanks again!

---

### Official Review · Reviewer_ZcuG · 2024-11-03

**Soundness:** 3
**Presentation:** 3
**Contribution:** 3
**Rating:** 6
**Confidence:** 3

**Summary:**

This paper introduces VideoAgent, an approach that refines video plans generated by video diffusion models through self-conditioning consistency and feedback from pre-trained vision-language models. When online interaction is available, VideoAgent closes the self-improvement loop by alternating between collecting successful online data and finetuning video models. Evaluations are performed on three benchmarks, across both simulated and real-world domains, and from the perspective of both task success rate and video generation quality, showing the effectiveness of the proposed method.

**Strengths:**

1. This paper proposes several novel techniques, such as self-conditioning consistency and incorporating VLM feedback, that improve the quality of the generated video plans through iterative refinement
2. The proposed method adopts a self-improving loop by finetuning the video models with additional successful trajectories collected online
3. The authors provide extensive evaluation results as well as experiment details, which back up the efficacy of the proposed method
4. This paper is well-motivated and easy to follow, it thoroughly discusses the limitations of the proposed work

**Weaknesses:**

1. The loop of collecting successful data through environment interaction and finetuning video models might incur large computational overhead, and the improvement seems to become marginal after two online iterations in Figure 4.
2. [nitpick] I believe it is essential to reduce hallucinations and improve the quality of video plans for better decision-making performance and I appreciate the overall contribution. However, in the Metaworld example (Table 1) provided in this work, simply replanning during inference still seems to be a comparatively cost-effective approach that can achieve higher overall success rates with a less performant video plan generator. I understand refinement and replanning can be combined for better improvement, but I wonder if the authors have stronger examples to highlight the necessity of refinement and self-improvement for task success.

**Questions:**

1. There seems to be a discrepancy between the self-conditioning-consistency loss In Algorithm 1 and Equation 7, is the second term in Eq. 7 missing in Algorithm 1?
2. During evaluation, is every video plan executed in an open-loop manner? What is the planning horizon for each setup, and how does it compare to the task horizon?
3. In Figure 4, how many additional trajectories are collected per iteration to achieve the improvement?
4. In the last row of Table 5, how is the "task success" defined specifically in the Bridge human evaluation? Is it defined as whether the generated videos respect the prompt from human perception? or as the authors mentioned in line 467, it depends on whether a generated video looks realistic.
5. As the video models are language-conditioned, it would be clearer to include the conditioning variable in the equations or algorithms somewhere.
6. Typo in line 342: brining

---

> ### Author Response · Authors · 2024-11-23
>
> Thank you for recognizing the significance of this work! Please see our response to your questions below.
> # W1. Computational overhead, marginal gains after two iterations (Figure 4)
> Robotic control from videos, as demonstrated by AVDC, is an exceptionally challenging problem with high relevance due to its potential for leveraging internet-scale knowledge transfer. While this task remains difficult, our method demonstrates significant improvements in specific tasks, including door-open, button-press, and faucet-close, which represent meaningful progress compared to prior work. Notably, little work has demonstrated improvements over AVDC, highlighting the significance of our results.
>
> Additionally, both The bitter lesson[1] and recent advancements in large language models (LLMs)[2] have shown that trading computational overhead for better performance has lots of potential. Our approach follows this principle, prioritizing performance gains while leveraging additional computational resources.
>
> # W2. Necessity of Refinement and Self-Improvement for Task Success
> We appreciate the reviewer’s valuable point about balancing refinement with replanning for cost-effective success. Refinement proves advantageous in precision-dependent tasks (e.g., 'faucet-close' in MetaWorld), where compounding errors from early frame inaccuracies could require frequent resets with replanning alone.
> a) **Offline Context:** In offline settings, where replanning is not feasible, refinement is essential for iteratively enhancing video plans without task-specific reinitialization.
>
> b) **Broad Applicability and Generalizability:** Refinement allows VideoAgent to integrate data and insights from other tasks (e.g., from refining human videos), creating broadly shareable improvements across tasks, embodiments, and potentially even human settings. Replanning, by contrast, is inherently task-specific, making refinement a more versatile and reusable enhancement strategy.
>
> c) The contribution of this work goes beyond better task success. A main contribution of this work is using robotic tasks to ground and improve video generation.
>
> # Q1. Missing Term in Algorithm 1 from Eqn 7
> We would like to clarify that Equation 7 and Algorithm 1 represent the same process. Equation 7 does not depict the standard diffusion loss but instead reflects the **self-consistency loss**, with both terms emphasizing self-consistency. During training, we set $\lambda = 0$. The consistency mechanism in VideoAgent is captured in the second term of Equation 7, where the loss function minimizes differences between videos generated from two individual samples at different timesteps. This mechanism promotes frame-to-frame coherence, enabling VideoAgent to iteratively refine its outputs.
>
> To address any potential confusion, we have updated the paper to explicitly state this connection and clarify the role of self-consistency in our method. Thank you for pointing this out and helping us improve the presentation of this concept.
>
> # Q2. Open-Loop Execution and Planning Horizon
> Each video plan is executed in an open-loop manner. The plan is first passed through a flow model, which processes it to determine grasp positions. Each video plan typically corresponds to approximately 60–100 time steps. The task horizon is set to a maximum length of 500 steps.
>
> # Q3. Additional Trajectories per Iteration
> To achieve this improvement, we collect **15 successful trajectories** for each task during every iteration and perform online finetuning on them. This standardization helps address task imbalance, as task success rates are higher for certain tasks compared to others. By ensuring a fixed number of successful trajectories per task, we prevent overfitting to easier tasks and maintain balanced model performance across the entire task set. We have incorporated this into the paper to clarify how we mitigate task imbalance during finetuning.
>
> # Q4. Definition of Task Success in Table 5
> In Table 5, "task success" measures only the proper completion of the task as defined by the initial goal-conditioning textual prompt, ignoring the relative differences in video generation quality. In Table 8, presented in the Appendix, we report both this task completion success rate and additional human evaluation metrics on fine-grained aspects such as video quality, temporal consistency, and more.
>
> # Q5. Include Conditioning Variable for Clarity
> Thank you for the suggestion. We have detailed how we condition on feedback in Equation 10 and in Section 3.1. We will ensure this is explicitly incorporated into Algorithm 2.
>
> # Q6. Typo in line 342
> Thank you for pointing this out. We have corrected this in the updated paper.
>
> [1] Sutton, Richard. "The bitter lesson." Incomplete Ideas (blog) 13.1 (2019): 38.
> [2] Snell, Charlie, et al. "Scaling llm test-time compute optimally can be more effective than scaling model parameters." arXiv preprint arXiv:2408.03314 (2024).

---

> ### Comment · Reviewer_ZcuG · 2024-11-26
> **Rebuttal Acknowledgement**
>
> Thank you to the authors for their efforts in addressing my concerns. Revised section 3.1 improves the clarity of the proposed method, and most of my questions have been addressed. I will maintain my positive rating. However, I do have a few suggestions for further improvement on the paper:
>
> - Comparing the "AVDC" and "VideoAgent" rows in Table 1, apart from "door-open" and "faucet-close," which clearly benefit from the refinement process, performance on some tasks (e.g., basketball, faucet-open, handle-press, and assembly) has also decreased. I believe including the standard deviation of these numbers may better clarify the performance variance.
>
> - In Section 3.1, the same $\lambda$ was overloaded for both Eq. 7 and Eq.9, it would be clearer to use different symbols to denote the coefficients

---

> > ### Author Response · Authors · 2024-12-01
> >
> > ## 1) Performance variance in AVDC vs. VideoAgent
> > Thank you for your observation regarding task performance and the suggestion to include standard deviations. We agree that tasks like "basketball," "faucet-open," "handle-press," and "assembly" are inherently challenging, as they involve precise gripping and object placement. These tasks performed poorly in the baseline (AVDC) itself, indicating their difficulty even before refinement. While VideoAgent’s refinement process improves the overall video generation quality, these specific tasks remain challenging due to their reliance on fine-grained motor actions and precise object interactions.
> >
> > We recognize the value of including standard deviations to better understand performance variance across seeds. We will include these in the updated version of the paper to clarify task variability and robustness. Despite these challenges, VideoAgent demonstrates significant improvements in overall task success rates and excels in tasks like "door-open" and "faucet-close," showcasing the effectiveness of the refinement process for long-horizon tasks.
> >
> >
> > ## 2) Clarify coefficients overloaded in section 3.1
> > To address this, we will revise the paper by assigning distinct symbols to the coefficients in these equations to clearly differentiate their roles. This update will improve clarity and ensure the equations are easier to interpret.
> >
> > We appreciate your careful review and constructive suggestion, which will help enhance the readability of our work.

---

### Meta-Review · Area_Chair_Y8Y5 · 2024-12-19

**Metareview:**

This paper proposes VideoAgent, a method for self-improving video generation. It refines video plans generated by video diffusion models through self-conditioning consistency and feedback from pre-trained vision-language models (VLMs). The refined video plans can be converted to robot actions, and successful trajectories are used for further fine-tuning. Experiments are conducted in simulated robotic manipulation and on real-robot videos, showing reduced hallucination and improved task success rates.

The paper has several novel aspects and the authors have made efforts to address the reviewers' concerns. However, the overall improvement from the proposed key modules (self-conditioning consistency, VLM feedback, real-world interaction feedback) is relatively small compared to the previous approach, which works across different video generation methods. Additionally, the lack of quantitative results for video generation quality makes it difficult to fully assess the effectiveness of the proposed approach. Despite the authors' explanations and plans for future work, these limitations are significant enough to warrant a rejection at this stage.

**Additional Comments On Reviewer Discussion:**

1. **Computational Overhead and Marginal Gains **:
    - **Point Raised**: Concern about the computational overhead of the loop collecting successful data and finetuning video models, and that the improvement seems marginal after two online iterations.
    - **Author's Response**: Emphasized the significance of the improvements in specific tasks compared to prior work, and that trading computational overhead for better performance has potential as shown by other works. Also noted the importance of the refinement process in precision-dependent tasks and its broader applicability.
    - **Weighing in Decision**: While the authors provided a reasonable explanation, the relatively small improvement in some tasks and the associated computational cost still raise concerns about the practicality and scalability of the proposed method.
2. **Necessity of Refinement **:
    - **Point Raised**: Questioned the necessity of refinement and self-improvement for task success, suggesting that replanning during inference might be a more cost-effective approach.
    - **Author's Response**: Highlighted the advantage of refinement in precision-dependent tasks where compounding errors could require frequent resets with replanning alone. Also mentioned the importance of refinement in offline settings and its ability to integrate data from other tasks, and that the contribution goes beyond just task success to grounding and improving video generation.
    - **Weighing in Decision**: The authors' response makes a valid point about the potential benefits of refinement in certain scenarios, but the overall impact on task success compared to simpler approaches like replanning is still a point of consideration.
3. **Consistency Model vs. DDPM **:
    - **Point Raised**: Unclear about the motivation for using the consistency model and its difference from DDPM and DDIM, as well as confusion about the loss function and the multi-step refinement process.
    - **Author's Response**: Clarified that the iterative refinement process in VideoAgent serves a similar purpose to traditional consistency models by aligning flawed video frames. Explained that the consistency mechanism allows for self-correction and progressive improvement, which DDPM and DDIM alone cannot achieve. Also detailed the loss function and the role of consistency in the refinement process.
    - **Weighing in Decision**: The authors' response helped clarify the concepts and the unique contribution of the consistency model, but the initial confusion raised questions about the clarity of the paper's presentation.
4. **Real-World Robot Manipulation Experiments **:
    - **Point Raised**: Recommended conducting experiments on a real robot and using more realistic benchmarks like Simper.
    - **Author's Response**: Focused on zero-shot video generation in SimplerEnv and plans to train an inverse dynamics model in future to validate task execution. Also provided examples of generated video plans in Appendix H1.
    - **Weighing in Decision**: While the future plans are promising, the lack of actual real-robot experiments and the current limitations in the presented results from SimplerEnv (such as blurry videos and unclear zero-shot generalization)削弱 the paper's practical relevance.

### Overall Decision
Although the authors made efforts to address the reviewers' concerns, several significant issues remained. The relatively small improvement from the proposed key modules compared to the Online-and-Replan approach, the lack of quantitative results for video generation quality, the limited real-robot experiments, and the initial confusion in some key concepts and presentations all contributed to the decision.

---

### Decision · Program_Chairs · 2025-01-22

Reject